# Learning Robust Hierarchical Patterns of Human Brain across Many fMRI Studies

**Dushyant Sahoo**
Department of Electrical Engineering
University of Pennsylvania
sadu@seas.upenn.edu

**Christos Davatzikos**
Department of Radiology
University of Pennsylvania
christos.davatzikos@uphs.upenn.edu

## Abstract

Multi-site fMRI studies face the challenge that the pooling introduces systematic non-biological site-specific variance due to hardware, software, and environment. In this paper, we propose to reduce site-specific variance in the estimation of hierarchical Sparsity Connectivity Patterns (hSCPs) in fMRI data via a simple yet effective matrix factorization while preserving biologically relevant variations. Our method leverages unsupervised adversarial learning to improve the reproducibility of the components. Experiments on simulated datasets display that the proposed method can estimate components with higher accuracy and reproducibility, while preserving age-related variation on a multi-center clinical data set.

## 1 Introduction

Multi-site fMRI studies have gained a lot of interest over the last decade [1, 2]. One reason for this is the necessity to evaluate a hypothesis in multiple settings/sites and make the hypothesis result generalizable to a diverse population. Also, the pooling of data is essential when studying rare disorders or neurological conditions where the aim is to generalize the results to diverse populations [3, 4]. However, the data pooling often results in the introduction of non-biological systematic variance due to differences in scanner hardware and imaging acquisition parameters [5]. This additional variability can lead to spurious results and a decrease in statistical power. The variability can also result in hindrance in the estimation of true biological changes or in inferring non-biological differences as biological because of the correlation between site effects and biological predictors. Many studies working with multi-site data fMRI have reported considerable variability due site or scanner effects [6, 7, 1].

The non-biological variability introduced due to inter-scanner and inter-protocol variations can affect the estimation of the common features derived from fMRI [8], such as functional connectivity [5] or sparse hierarchical factors (this paper). These features were used for the study of the brain's function during aging [9], of various neurological disorders [10, 11], and tasks [12]. The non-biological variability can considerably reduce these features' reproducibility across different datasets and hence their utility as biomarkers for diseases that disrupt functional connectivity. Thus the removal of non-biological variance introduced by pooling of the data is essential for many neuroimaging studies.

Independent Component Analysis [13], Non Negative Matrix Factorization [14], Sparse Dictionary Learning [15] and Sparse factor Modeling [16, 17] are some of the commonly used methods for estimating interpretable components that capture the complexity of the human brain function at rest. However, these methods do not account for the hierarchical organization of the human brain [18, 19]. Recently, deep learning based methods [20, 21] have been developed to estimate hierarchical networks. However, they fail to capture overlapping components and the existence of positively and negatively correlated nodes in a component that are shown to capture the underlying structure better [22, 23]. Hierarchical Sparse Connectivity Patterns (hSCPs) overcome these limitations by

capturing interpretable hierarchical, sparse, and overlapping components using a linear deep matrix factorization approach. The method not only captures shared patterns but also the subject-specific weights of these patterns, thus capturing the heterogeneity of brain activity patterns across individuals. In this paper, we focus on robust estimation of hSCPs [23–25] in a multi-site regime.

Many existing methods to reduce site effects are based on an empirical Bayes method ComBat [26], which was developed to remove 'batch effects' in genetics and has been applied for harmonizing different measures derived from structural [27, 28] and functional MRI [8]. But these harmonization methods can not be directly applied to the hSCP method because of the loss in the structure of captured heterogeneity (more details in section 2.1). See Appendix A for extended discussion on related work.

**Contribution.** In this paper, we develop a new model that is robust to site-effects in the estimation of sparse hierarchical connectivity pattern components (rshSCP). For this, the method learns site-specific features and global space, storing the information about the scanner and site, and uses these features to reduced site effects in the components. We also use an existing adversarial learning approach [25] on top of our method to improve the reproducibility and generalizability of the components across components from the same site. We formulate the method as a non-convex optimization problem which is solved using stochastic gradient descent. Experiments on simulated and real datasets show that our method can improve the split-sample and leave one site reproducibility of the components while retaining age-related biological variability in the data, thus capturing informative heterogeneity.

## 2 Method

In this section, we first present a brief overview of the hierarchical Sparse Connectivity Pattern (hSCP) and its adversarial formulation. We then describe our main method and the joint formulation incorporating adversarial learning. We follow the notation in [24] and [25]. The set of symmetric positive definite matrices of size $P \times P$ is denoted by $\mathbb{S}_{++}^{P \times P}$. Matrix $\mathbf{A}$ with all the elements greater than or equal to 0 is denoted by $\mathbf{A} \geq 0$. $\mathbf{J}_P$ denotes $P \times P$ matrix with all elements equal to one. $P \times P$ identity matrix is denoted by $\mathbf{I}_P$ and element-wise product between two matrices $\mathbf{A}$ and $\mathbf{B}$ is denoted by $\mathbf{A} \circ \mathbf{B}$.

### 2.1 Introduction to hierarchical Sparse Connectivity Patterns

Hierarchical Sparse Connectivity Patterns (hSCP) [24] is a hierarchical extension of Sparse Connectivity Patterns (SCPs), first defined by [23] to estimate sparse functional patterns in the human brain using fMRI data. Let there be $N$ number of subjects or participants, and each subject's BOLD fMRI time series has $T$ time points and $P$ nodes representing regions of interest. The input to hSCP are correlation matrices $\mathbf{\Theta}^n \in \mathbb{S}_{++}^{P \times P}$ where $i$th and $j$th element of the matrix is the correlation between time series of $i$th and $j$th node. hSCP then outputs a set of shared hierarchical patterns following the below equations:

$$\mathbf{\Theta}^n \approx \mathbf{W}_1 \mathbf{\Lambda}_1^n \mathbf{W}_1^\top, \quad \ldots \quad \mathbf{\Theta}^n \approx \mathbf{W}_1 \mathbf{W}_2 \ldots \mathbf{W}_K \mathbf{\Lambda}_K^n \mathbf{W}_K^\top \mathbf{W}_{K-1}^\top \ldots \mathbf{W}_1^\top,$$

where $\mathbf{\Lambda}_k^n$ is a diagonal matrix having positive elements storing relative contribution of the components for the $n$th subject at $k$th level, $K$ is the depth of hierarchy and $P > k_1 > \ldots > k_K$ i.e. each successive level in the hierarchy has less number of components than the previous one. In the above formulation, $\mathbf{W}_1 \in \mathbb{R}^{P \times k_1}$ stores $k_1$ components at the bottom most level, and each successive multiplication by $\mathbf{W}_2, \mathbf{W}_3, \ldots, \mathbf{W}_K$ linearly transforms to a lower dimensional space of $k_2, k_3, \ldots, k_K$ dimension. Let $\mathcal{W} = \{\mathbf{W}_r \mid r = 1, \ldots, K\}$ be the set storing sparse components shared across all subjects and $\mathcal{D} = \{\mathbf{\Lambda}_r^n \mid r = 1, \ldots, K; n = 1 \ldots, N\}$ be set storing subject specific diagonal matrix with $\mathbf{\Lambda}_r^n \geq 0$. The hierarchical components are estimated by solving the below optimization problem:

$$\min_{\mathcal{W}, \mathcal{D}} \quad H(\mathcal{W}, \mathcal{D}, \mathcal{C}) = \sum_{n=1}^{N} \sum_{r=1}^{K} \|\mathbf{\Theta}^n - (\prod_{j=1}^{r} \mathbf{W}_j) \mathbf{\Lambda}_r^n (\prod_{j=1}^{r} \mathbf{W}_j)^\top\|_F^2$$

$$\text{s.t.} \quad \|\mathbf{w}_l^r\|_1 < \tau_r, \ \|\mathbf{w}_l^r\|_\infty \leq 1, \ \text{trace}(\mathbf{\Lambda}_r^n) = 1,$$
$$\mathbf{\Lambda}_r^n \geq 0, \ \forall n, r, l; \qquad \mathbf{W}_j \geq 0, \ j = 2, \ldots, K,$$

(1)

where $\mathcal{C} = \{\boldsymbol{\Theta}^n \mid n = 1, \ldots, N\}$, $l = 1, \ldots, k_r$, $n = 1, \ldots, N$ and $r = 1, \ldots, K$. $L_1$, $L_\infty$ and trace constraints help the problem to identify a decomposition which can provide reproducible components. More details can be found in the original paper [24]. We will be denoting above constraint set as $\Omega_\mathcal{W} = \{\mathbf{W} \mid \|\mathbf{w}_l^r\|_1 < \tau_r, \|\mathbf{w}_l^r\|_\infty \leq 1, \mathbf{W}_j \geq 0, j = 2, \ldots, K\}$ and $\Psi = \{\boldsymbol{\Lambda} \mid \mathrm{trace}(\boldsymbol{\Lambda}_r^n) = 1, \boldsymbol{\Lambda}_r^n \geq 0\}$. A detailed description of the method can be found in [24].

**Can we use standard harmonization approaches?** These methods reduce site effects by adjusting for additive and multiplicative effects for each feature in data separately and use emperical Bayes estimates the model parameters. These methods can be used in the case of hSCP in two ways. First, harmonization can be directly applied to each element of the correlation matrices, which is the input of hSCP. This will reduce site effects from each element of the correlation matrix, thus from the complete input, but the final matrix that does not necessarily follow the essential property of a correlation matrix i.e., positive definiteness. For similar reasons, COMBAT can not be directly applied to time series; if applied, it can change the inference derived from the correlation matrix. Second, harmonization can be directly applied to $\boldsymbol{\Lambda}$ to remove site effects. To understand this, we look at the hSCP formulation at one level:

$$\boldsymbol{\Theta}^n \approx \sum_{l=1}^{k} d_l^n \mathbf{w}_l \mathbf{w}_l^\top \approx \mathbf{W} \boldsymbol{\Lambda}^n \mathbf{W}^\top,$$

where $d_l^n$ are non-zero elements storing the subject-specific information, which can be affected by the variability introduced by the site. In this model, harmonizing each feature across different sites will change the relative contribution of the components in each subject's functional structure, which will remove the interpretability of the subject-wise weights, which is not desirable. Instead, a two step optimization procedure can be used to incorporate ComBat with hSCP (ComBat hSCP). We first run hSCP and use ComBat on the extracted $\boldsymbol{\Lambda}^n$ to get harmonized subject specific information $\boldsymbol{\Delta}^n \in \mathbb{R}^{k_1 \times k_1}$ for each subject. We then re-fitted $\mathbf{W}$ using the below decomposition-

$$\boldsymbol{\Theta}^n \approx \mathbf{W}(\boldsymbol{\Delta}^n + \mathbf{S})\mathbf{W}^\top.$$

We added a diagonal shift matrix $\mathbf{S} \in \mathbb{R}^{k_1 \times k_1}$ such that $\boldsymbol{\Delta}^n + \mathbf{S}$ is positive for each subject and performed the optimization to estimate $\mathbf{W}$ and $\mathbf{S}$. We show through experiments that this baseline two step optimization procedure is not optimal and performs worse than vanilla hSCP.

## 2.2 Adversarial Learning in hSCP

Adversarial learning has shown to achieve state-of-the-art performance of various matrix factorization approaches [29, 30]. Recently, Sahoo and Davatzikos [25] demonstrated that incorporating adversarial learning in the estimation problem of hSCP can improve the reproducibility of the hierarchical components. The method is based on perturbation of input data $\boldsymbol{\Theta}^n$ to learn stable components robust to adversaries. First the input data is perturbed to generate new data $\boldsymbol{\Gamma}^n = \boldsymbol{\Theta}^n + \sigma \mathbf{J}_P$ where $\sigma$ is the standard deviation of the data. We consider this perturbation as rank one perturbation which will transform eigenvalues of each subject's correlation matrix differently. Since the addition of ones does not change the symmetric property of the matrix, we just scale the matrix such that the diagonal matrix contains 1 and by Weyl's inequality about perturbation [31] it will be positive definite. We have experimented with randomly positively scaled rank one perturbation, but the results were worse than the vanilla hSCP. Also, using rank one perturbation, it is easier to control eigenvalues (and noise added) of the modified matrix than using rank k perturbation. The perturbed set of components $\tilde{\mathbf{W}}_1$ are then estimated using the new data and are ensured to be close to $\mathbf{W}_1$ by solving the below minimization problem:

$$A(\hat{\mathbf{W}}_1) = \alpha \|\tilde{\mathbf{W}}_1 - \mathbf{W}_1\|_F^2 + \sum_{n=1}^{N} \|\boldsymbol{\Gamma}^n - \tilde{\mathbf{W}}_1 \boldsymbol{\Lambda}^n \tilde{\mathbf{W}}_1^\top\|_F^2. \tag{2}$$

In the above optimization problem, the first term constraints $\tilde{\mathbf{W}}_1$ to be close to $\mathbf{W}_1$ and second term is used for learning the components using the perturbed data. Aim of the learner is to estimate $\boldsymbol{\Lambda}^n$ and $\mathbf{W}_1$ by minimizing the below cost function:

$$D(\mathbf{W}, \boldsymbol{\Lambda}) = \sum_{n=1}^{N} \|\boldsymbol{\Theta}^n - \tilde{\mathbf{W}}_1 \boldsymbol{\Lambda}^n \tilde{\mathbf{W}}_1^\top\|_F^2 + \beta \sum_{n=1}^{N} \|\boldsymbol{\Theta}^n - \mathbf{W}_1 \boldsymbol{\Lambda}^n \mathbf{W}_1^\top\|_F^2, \tag{3}$$

for a fixed $\tilde{\mathbf{W}}_1$. Learner first estimates subject specific information $\mathbf{\Lambda}^n$ using perturbed weight matrix and then use it to learn $\mathbf{W}_1$. Below is the complete optimization problem for adversarial learning of hSCP:

$$\min_{\mathcal{W},\mathcal{D}} \quad H(\tilde{\mathcal{W}}, \mathcal{D}, \mathcal{C}) + \beta H(\mathcal{W}, \mathcal{D}, \mathcal{C})$$

$$\text{s.t.} \quad \tilde{\mathbf{W}}_r = \underset{\hat{\mathbf{W}}_r}{\arg\min} \, \alpha \|\hat{\mathbf{W}}_r - \mathbf{W}_r\|_F^2 + H(\tilde{\mathcal{W}}, \mathcal{D}, \mathcal{P}) \quad r = 1, \dots, K \tag{4}$$

$$\tilde{\mathbf{W}}_r, \mathbf{W}_r \in \Omega, \qquad \mathcal{D} \in \Psi,$$

where $\tilde{\mathcal{W}} = \{\tilde{\mathbf{W}}_r \mid r = 1, \dots, K\}$ and $\mathcal{P} = \{\mathbf{\Gamma}^n \mid n = 1, \dots, N\}$. We next discuss our main method which aims to reduce the site effects.

## 2.3  Robust to site hSCP

Estimating hSCPs in multi-site data can introduce non-biological variances in the components and the subject-specific information. One of the typical approaches would be to use harmonization methods mentioned previously, but it would lead to a loss in the structure of these features, which in turn will lose interpretability. Instead of removing site effects after estimating the components, we jointly model the sparse components and the site effects, and estimate robust to site hSCP (rshSCP). We first look at the case when there is only one level in the hierarchy, which then can be extended to multiple levels. Let there be total $S$ sites, $\mathcal{I}_s$ be the set storing subjects from site $s$ and $\mathbf{y} \in \mathbb{R}^{N \times S}$ be the one-hot encoded site labels We hypothesize that there is a space $\mathbf{V} \in \mathbb{R}^{P \times P}$ storing site and scanner information for all the possible available data, and for each site $s$, we have space $\mathbf{U}^s \in \mathbb{R}^{P \times P}$ storing site-specific information for $s = 1, \dots, S$. Based on the above hypothesis, we decompose the correlation matrix $\mathbf{\Theta}^n$ of $n \in \mathcal{I}_s$ to jointly estimate the hSCPs, $\mathbf{U}^s$ and $\mathbf{V}$ as:

$$\mathbf{\Theta}^n \approx \underbrace{\mathbf{W}\mathbf{\Lambda}^n\mathbf{W}^\top}_{\substack{\text{decomposition of} \\ \text{subject components}}} + \underbrace{\mathbf{U}^s\mathbf{V}}_{\substack{\text{decomposition of} \\ \text{site components}}}, \tag{5}$$

where $\mathbf{U}^s$ is constrained to be a diagonal matrix and $L_1$ sparsity constraint is used for $\mathbf{V}$ to prevent overfitting. In addition to estimating the site effects, we train the model such that the predictive power of subject-specific information $\mathcal{D}$ for predicting site is reduced, which can assist in removing site information. For this, we train a differentiable classification model $F(\zeta, \mathcal{D})$ parameterized by $\zeta$ with input $\mathbf{\Lambda}^n$ that return site predictions $\hat{\mathbf{y}} \in \mathbb{R}^{N \times S}$. These predictions indicate the probabilities that each of $N$ inputs belongs to each of $S$ site labels. The classification model is trained by optimizing for $\zeta$ such that the cross-entropy loss $\mathcal{L}(\zeta, \mathcal{D}, \mathbf{y})$ between the predictions $\hat{\mathbf{y}}$ and the true site labels $\mathbf{y}$ is minimized:

$$\zeta^* = \underset{\zeta}{\arg\min} -\frac{1}{N} \sum_{n=1}^{N} \sum_{s=1}^{S} y_{n,s} \log \hat{y}_{n,s}. \tag{6}$$

Using this model, we modify $\mathbf{\Lambda}^n$ such that its predictability power reduces. We achieve this by maximizing the above loss with respect to $\mathcal{D}$. This will result in a minimax game, where the classifier learns to minimize the cross-entropy or the surrogate classification loss, and $\mathcal{D}$ is adjusted to maximize the loss. The joint optimization problem can be written as:

$$\max_{\zeta} \min_{\mathbf{W},\mathcal{D},\mathcal{U},\mathbf{V}} \quad \sum_{s=1}^{S} \sum_{n \in \mathcal{I}_s} \|\mathbf{\Theta}^n - \mathbf{W}\mathbf{\Lambda}^n\mathbf{W}^\top - \mathbf{U}^s\mathbf{V}\|_F^2 - \gamma\mathcal{L}(\zeta, \mathcal{D}, \mathbf{y}) \tag{7}$$

$$s.t. \quad \mathbf{W} \in \Omega, \quad \mathcal{D} \in \Psi, \quad \|\mathbf{v}_p\|_1 < \mu, \, p = 1, \dots, P,$$

where $\mathcal{U} = \{U_s \mid s = 1, \dots, S\}$, $\mathbf{v}_p$ is the $p$th column of $\mathbf{V}$.

## 2.4  Complete Model

We can combine the above formulation (7) at multi level with the adversarial learning (4) to jointly model hSCPS and site effects. Let

$$G(\mathcal{W}, \mathcal{D}, \mathcal{C}) = \sum_{s=1}^{S} \sum_{n \in \mathcal{I}_s} \sum_{r=1}^{K} \|\mathbf{\Theta}^n - (\prod_{j=1}^{r} \mathbf{W}_j)\mathbf{\Lambda}_r^n(\prod_{n=1}^{r} \mathbf{W}_n)^\top - \mathbf{U}_r^s\mathbf{V}_r\|_F^2, \tag{8}$$

then the joint optimization problem can be written as:

$$\max_{\zeta} \min_{\mathcal{W},\mathcal{D},\mathcal{U},\mathbf{V}} \quad J(\tilde{\mathcal{W}},\mathcal{W},\mathcal{D},\mathcal{C}) = G(\tilde{\mathcal{W}},\mathcal{D},\mathcal{C}) + \beta G(\mathcal{W},\mathcal{D},\mathcal{C}) + \gamma \mathcal{L}(\zeta,\mathcal{D},\mathbf{y})$$

$$\text{s.t.} \quad \tilde{\mathbf{W}}_r = \arg\min_{\hat{\mathbf{W}}_r} \alpha\|\hat{\mathbf{W}}_r - \mathbf{W}_r\|_F^2 + G(\tilde{\mathcal{W}},\mathcal{D},\mathcal{P}) \quad r = 1,\dots,K \qquad (9)$$

$$\tilde{\mathbf{W}}_r, \mathbf{W}_r \in \Omega \quad \mathcal{D} \in \Psi, \quad \|\mathbf{v}_p\|_1 < \mu, \; p = 1,\dots,P,$$

The optimization problem defined above is a non-convex problem that we solve using alternating minimization. Complete algorithm and the details about the optimization are described in Appendix B. Note that the random initialization of the variables can result in a very different final solution that might be far from the ground truth. One such solution for $\mathbf{U}$ and $\mathbf{V}$ would be the identity matrix since all the correlation matrices have one as their diagonal element, which can drastically change the final components. It might also be possible that $\mathbf{V}$ might store highly reproducible components since they are present in most individuals, leading to a decrease in reproducibility of hSCPs. We prevent these cases by using $\mathrm{svd-initialization}$ [24, Algorithm 2] for $\mathcal{W}$ and $\mathcal{D}$, where, in the starting, most of the variability associated with data is stored in $\mathcal{W}$ and $\mathcal{D}$. In this way, we can prevent $\mathbf{V}$ from storing highly reproducible components during initial iterations. We initialize $\mathbf{U}^s$ and $\mathbf{V}$ using the below equation:

$$\mathbf{U}_r^s = \left[\frac{1}{|\mathcal{I}_s|}\left(\sum_{n\in\mathcal{I}_s}\mathbf{\Theta}^n - (\prod_{j=1}^{r}\mathbf{W}_j)\mathbf{\Lambda}_r^n(\prod_{n=1}^{r}\mathbf{W}_n)^\top\right)\mathbf{J}_p\right] \circ \mathbf{I}_p, \qquad \mathbf{V}_r = \frac{1}{P}\mathbf{J}_P. \qquad (10)$$

This complete initialization procedure ensures that the algorithm starts with the majority of variability in the data stored in $\mathcal{W}$ and $\mathcal{D}$, and $\mathbf{U}^s$ start from the residual variance left in site $s$ after the $\mathrm{svd-initialization}$ procedure. We show in the next sections that this simple strategy, though suboptimal, can help estimate reproducible components with diminished site effects. All the code is implemented in MATLAB and will be released upon publication.

# 3 Experiment

## 3.1 Simulated Dataset

**One level.** We first generate simulated dataset at one level to evaluate the performance of our model against the vanilla hSCP. We simulate data with $p = 50$, $k_1 = 10$, $S = 4$ with 200, 300, 400 and 500 number of participants in each site. We generated sparse shared components $\mathbf{W}_1$ with percentage of non-zeros equal to $60\%$ and each element sampled from $\mathcal{N}(0,1)$. We then generate correlation matrix for $n$th subject belonging to $s$th site using:

$$\mathbf{\Theta}^n = \left(\mathbf{W}_1 + \mathbf{E}_1^n\right)\mathbf{\Lambda}^n\left(\mathbf{W}_1 + \mathbf{E}_2^n\right)^\top + \mathbf{U}^s\mathbf{V} + \mathbf{E}_2^n, \qquad (11)$$

where $\mathbf{U}^s$ is a diagonal matrix with positive elements sampled from $\mathcal{N}(1,.1)$, $\mathbf{V}$ is a random matrix sampled from wishart distribution, each element of $\mathbf{\Lambda}^n$ is sampled from $\mathcal{N}(4,1)$ and $\mathbf{E}_1^n$ is the noise matrix added to the components whose each element is sampled from $\mathcal{N}(0,.1)$ and $\mathbf{E}_2^n$ is added to ensure that the final matrix is positive definite. However the diagonal elements of $\mathbf{\Theta}^n$ are not equal to 1. To make them 1, we extract diagonal elements $\mathbf{D}$ of $\mathbf{\Theta}^n$ and get the new correlation matrix as $\mathbf{D}^{1/2}\mathbf{\Theta}^n\mathbf{D}^{1/2}$. We used a feed-forward neural network for the classification model with two hidden layers. The networks contain the following layers: a fully connected layer with 50 hidden unites, dropout layer with rate 0.2, ReLU, a fully-connected layer with 4 hidden units and a softmax layer. Optimal value of hyperparameters $\alpha$, $\beta$, $\mu$ and $\tau_1$ are selected from $[0.1,1]$, $[1,5]$, $[0.1, 0.5, 1]$ and $10^{[-2:2]}$. The criterion for choosing the best hyperparameter is maximum split-sample reproducibility. The split sample reproducibility is the normalized inner product between the components estimated on two random equal splits of the data. Split sample reproducibility tries to answer the question of whether the components are generalizable across subjects from the same sites or not. We compared different methods for estimation of hierarhical components- hSCP, ComBat hSCP, hSCP with adversarial learning (Adv. hSCP), rshSCP, rshSCP with adversarial learning (Adv. rshSCP), rshSCP and Adv. rshSCP with random initialization (rshSCP w/ rand. and Adv. rshSCP w/ rand.). Table 2 shows the reproducibility of the components generated from different methods. It is computed over 15 runs in all the experiments. We used accuracy of the estimated components as a

Table 1: Accuracy of the components on simulated dataset at one level.

| Method | $k_1 = 8$ | $k_1 = 10$ | $k_1 = 12$ | $k_1 = 14$ |
|---|---|---|---|---|
| hSCP | 0.789 | 0.787 | 0.745 | 0.736 |
| ComBat hSCP | 0.763 | 0.759 | 0.731 | 0.718 |
| Adv. hSCP | 0.869 | 0.875 | 0.862 | 0.854 |
| rshSCP | 0.873 | 0.865 | 0.843 | 0.867 |
| Adv. rshCP | **0.903** | **0.910** | **0.902** | **0.908** |
| rshSCP w/ rand. | $0.856 \pm 0.039$ | $0.834 \pm 0.055$ | $0.824 \pm 0.031$ | $0.818 \pm 0.036$ |
| Adv. rshSCP. w/ rand. | $0.897 \pm 0.030$ | $0.895 \pm 0.036$ | $0.892 \pm 0.023$ | $0.886 \pm 0.034$ |

Table 2: Split sample reproducbility on simulated dataset at one level.

| Method | $k_1 = 8$ | $k_1 = 10$ | $k_1 = 12$ | $k_1 = 14$ |
|---|---|---|---|---|
| hSCP | $0.769 \pm 0.052$ | $0.798 \pm 0.047$ | $0.739 \pm 0.053$ | $0.734 \pm 0.047$ |
| ComBat hSCP | $0.749 \pm 0.040$ | $0.750 \pm 0.052$ | $0.724 \pm 0.049$ | $0.719 \pm 0.052$ |
| Adv. hSCP | $0.781 \pm 0.037$ | $0.818 \pm 0.031$ | $0.780 \pm 0.034$ | $0.750 \pm 0.031$ |
| rshSCP | $0.825 \pm 0.039$ | $0.845 \pm 0.030$ | $\mathbf{0.826 \pm 0.039}$ | $0.779 \pm 0.036$ |
| Adv. rshSCP | $\mathbf{0.840 \pm 0.044}$ | $\mathbf{0.869 \pm 0.034}$ | $0.815 \pm 0.035$ | $\mathbf{0.802 \pm 0.030}$ |
| rshSCP w/ rand. | $0.804 \pm 0.085$ | $0.818 \pm 0.086$ | $0.780 \pm 0.068$ | $0.758 \pm 0.071$ |
| Adv. rshSCP w/ rand. | $0.826 \pm 0.069$ | $0.833 \pm 0.081$ | $0.801 \pm 0.074$ | $0.782 \pm 0.077$ |

Table 3: Accuracy of the components on hierarchical simulated dataset.

| Method \ $k_1$ | $k_2 = 4$ | | | | $k_2 = 6$ | | | |
|---|---|---|---|---|---|---|---|---|
| | 8 | 10 | 12 | 14 | 8 | 10 | 12 | 14 |
| hSCP | 0.806 | 0.801 | 0.783 | 0.777 | 0.797 | 0.790 | 0.773 | 0.766 |
| ComBat hSCP | 0.788 | 0.776 | 0.743 | 0.729 | 0.779 | 0.766 | 0.747 | 0.734 |
| Adv. hSCP | 0.875 | 0.872 | 0.870 | 0.864 | 0.863 | 0.859 | 0.849 | 0.851 |
| rshSCP | 0.881 | 0.876 | 0.860 | 0.862 | 0.874 | 0.871 | 0.852 | 0.859 |
| Adv. rshSCP | **0.904** | **0.909** | **0.904** | **0.907** | **0.902** | **0.903** | **0.904** | **0.902** |

performance measure. It is defined as the normalized inner product between ground truth components and estimated components. All the experiments were run on a four i7-6700HQ CPU cores single ubuntu machine.

**Accuracy.** Table 1 displays the accuracy of different methods on the simulated dataset. Here, accuracy is defined as the average correlation between estimated components and the ground truth components. From the results, we can see that the rshSCP with adversarial learning can significantly improve the components' accuracy and the reproducibility of the components. The baseline (ComBat hSCP) performs worse than vanilla hSCP. One reason for this might be that the harmonized $\mathbf{\Lambda}$ extracted using ComBat might not necessarily result in optimal highly reproducible $\mathbf{W}$. This result bolsters our method that we need a joint optimization procedure to obtain $\mathbf{W}$ and $\mathbf{\Lambda}$ with reduced site effects. The results using random initialization instead of using the initialization strategy mentioned in the previous section indicates that random initialization brings significant variability to performance. On average, it performs worse than our strategy, but there might be instances where the random initialization can perform better, which might suggest that there might be some better strategy for initialization. Also, for $\mathbf{V}$, there is an optimal sparsity value, which achieves the best result. If $\mathbf{V}$ is dense, then it might remove essential information that might reduce reproducibility, and if it is too sparse, then we might not have desired effects to make the model robust. The results showing the variation in the accuracy and reproducibility with the sparsity of $\mathbf{V}$ are in Appendix D.1.

**Site prediction.** To check if the estimated subject information ($\mathbf{\Lambda}$) has reduced predictive power to predict the site to which the subject belonged, we performed a 5 fold cross-validation using SVM with RBF kernel. We also ran our experiment using two different feed forward networks with two different architectures: (a) a fully connected layer with hidden units, dropout layer with the rate

Table 4: Summary characteristics of the real dataset.

| Data Sites | Participants | % of Females | Age Range (Median) | Scanner |
|---|---|---|---|---|
| BLSA-3T | 784 | 56.5 | $[22, 95] (68)$ | 3T Philips |
| CARDIA1 | 199 | 55.7 | $[42, 61] (52)$ | 3T Siemens Tim Trio |
| CARDIA2 | 321 | 51.4 | $[43, 61] (52)$ | 3T Philips Achieva |
| CARDIA3 | 278 | 55.3 | $[43, 62] (52)$ | 3T Philips Achieva |
| UKBB | 2023 | 55.2 | $[45, 79] (63)$ | 3T Siemens Skyra |
| OASIS | 847 | 56.0 | $[42.6, 97] (70)$ | 1.5T Siemens Vision |
| ABC | 279 | 59.1 | $[23, 95] (70)$ | 3T Siemens Tim Trio |

, ReLU, a fully-connected layer with hidden units and a softmax layer and (b) a fully connected layer with hidden units, dropout layer with the rate , ReLU, a fully-connected layer with hidden units, dropout layer with rate , ReLU, a fully-connected layer with hidden units and a softmax layer. Our model leads to a decrease in average cross-validation accuracy from 97.6% to 67% for SVM, 98.1% to 67.3% for neural network with architecture (a) and 98.2% to 66.9% for neural network with architecture (b). This suggests that our model can reduce the prediction capability to predict site. More details about the result are in Appendix D.1.

**Two level.** Under the same settings as defined above, we generate correlation matrix from two level components with $k_2 = 4$ using:

$$\Theta^n = \tilde{\mathbf{W}}_1 \tilde{\mathbf{W}}_2 \mathbf{\Lambda}^n \tilde{\mathbf{W}}_2^\top \tilde{\mathbf{W}}_1^\top + \mathbf{U}^s \mathbf{V} + \mathbf{E}_3^n,$$
$$\tilde{\mathbf{W}}_1 = \mathbf{W}_1 \mathbf{E}_1^n, \quad \tilde{\mathbf{W}}_2 = \mathbf{W}_2 + \mathbf{E}_2^n \tag{12}$$

where each element of $\mathbf{W}_2$ is sampled from $\mathcal{N}(0, 1)$, the percentage of non-zeros equal to 40%, $\mathbf{E}_1^n$ and $\mathbf{E}_2^n$ is the noise added to the components whose each element is sampled from $\mathcal{N}(0, .1)$ and $\mathbf{E}_3^n$ is added to ensure that the final matrix is positive definite. Table 3 shows the accuracy for different values of $k_1$ and $k_2$. Selection of hyparameter is same as in the previous paragraph. We can see that the proposed method estimates most accurate ground truth components. Reproducbility results and random initialization results are availble in Appendix D.1.

## 3.2 Real Dataset

**Data.** We collected functional MRI data from 5 different multi-center imaging studies- 1) Baltimore Longitudinal Study of Aging (BLSA) [32, 33], the Coronary Artery Risk Development in Young Adults study (CARDIA) [34], UK BioBank (UKBB) [35], Open access series of imaging studies (OASIS) [36] and Aging Brain Cohort Study (ABC) from Penn Memory Center [37]. Although UK Biobank has more than 20000 scans, we only used 2023 randomly selected scans to avoid estimating the results that would be heavily influenced by the UK Biobank. We projected the data into a lower-dimensional space such that the number of nodes in each subject's data was 100. Table 4 summarizes the number of participants in each site and age distribution. More details on the dataset and the preprocessing pipeline are given in Appendix C. CARDIA data is divided into three parts because of the acquisition at three different sites.

**Reproducibility.** Since we don't have access to ground truth here, we compare the methods based on the split sample and leave one site reproducibility. Leave one site out reproducibility is defined as the similarity between components derived from the site $s$ and all sites except $s$. Split sample reproducibility tries to answer the question of whether the components are generalizable to other sites or not. For estimating rshSCP with only one site, we used $\mathbf{V}$ estimated from all sites except $s$ since the idea behind $\mathbf{V}$ was to store information about the site/scanner from various sites. This would also help analyze the generalization power of $\mathbf{V}$. The optimum value of the hyperparameters is selected from the range defined in section 3.1. $\tau_1$ and $\tau_2$ are selected from $10^{[-2:2]}$ based on maximum split-sample reproducibility. The criterion for choosing the best value is the maximum split sample reproducibility. Table 5 shows the split sample reproducibility for varied values of $k_1$ and $k_2 = 4$. Leave one site out reproducibility results are shown in Table 6. Table 13 and Table 14 in D.2 shows split sample reproducibility and leave one site out reproducibility respectively at two-level for $k_2 = 6$. The results demonstrate that the proposed method can significantly improve the split sample reproducibility and leave one site out reproducibility. For the remaining paper, we focus on

Table 5: Split-sample reproducbility on real dataset ($k_2 = 4$).

| Method | $k_1 = 10$ | $k_1 = 15$ | $k_1 = 20$ | $k_1 = 25$ |
|---|---|---|---|---|
| hSCP | $0.713 \pm 0.039$ | $0.707 \pm 0.038$ | $0.697 \pm 0.035$ | $0.683 \pm 0.036$ |
| ComBat hSCP | $0.673 \pm 0.049$ | $0.641 \pm 0.051$ | $0.639 \pm 0.031$ | $0.611 \pm 0.038$ |
| Adv. hSCP | $0.737 \pm 0.041$ | $0.719 \pm 0.033$ | $0.715 \pm 0.037$ | $0.710 \pm 0.043$ |
| rshSCP | $0.806 \pm 0.036$ | $0.768 \pm 0.032$ | $0.742 \pm 0.033$ | $0.743 \pm 0.044$ |
| Adv. rshSCP | $\mathbf{0.808 \pm 0.030}$ | $\mathbf{0.772 \pm 0.036}$ | $\mathbf{0.747 \pm 0.034}$ | $\mathbf{0.746 \pm 0.036}$ |

Table 6: Leave one site out reproducbility on real dataset($k_2 = 4$).

| Method | $k_1 = 10$ | $k_1 = 15$ | $k_1 = 20$ | $k_1 = 25$ |
|---|---|---|---|---|
| hSCP | $0.652 \pm 0.038$ | $0.618 \pm 0.041$ | $0.592 \pm 0.033$ | $0.571 \pm 0.035$ |
| ComBat hSCP | $0.614 \pm 0.042$ | $0.594 \pm 0.035$ | $0.542 \pm 0.041$ | $0.528 \pm 0.039$ |
| Adv. hSCP | $0.656 \pm 0.035$ | $0.629 \pm 0.039$ | $0.601 \pm 0.035$ | $0.584 \pm 0.034$ |
| rshSCP | $0.712 \pm 0.034$ | $0.701 \pm 0.036$ | $0.676 \pm 0.038$ | $0.665 \pm 0.034$ |
| Adv. rshSCP | $\mathbf{0.716 \pm 0.032}$ | $\mathbf{0.709 \pm 0.031}$ | $\mathbf{0.688 \pm 0.034}$ | $\mathbf{0.671 \pm 0.033}$ |

Table 7: Mean absolute error ($k_2 = 4$)

| Method | $k_1 = 10$ | $k_1 = 15$ | $k_1 = 20$ | $k_1 = 25$ |
|---|---|---|---|---|
| hSCP | $6.490 \pm 1.485$ | $6.468 \pm 1.442$ | $6.425 \pm 1.412$ | $6.414 \pm 1.417$ |
| Adv. rshSCP | $6.494 \pm 1.501$ | $6.467 \pm 1.475$ | $6.432 \pm 1.483$ | $6.409 \pm 1.470$ |

the comparison between components learned using adversarial learning from hSCP and Adv. rshSCP.

**Site prediction.** We performed the same experiment under the same settings as mentioned in the previous section to check $\mathbf{\Lambda}$ has reduced predictive power to predict the site. Using SVM, our model leads to a decrease in average cross-validation accuracy from $51\%$ to $32\%$. Using the first neural network architecture defined in section 3.1, the cross-validation accuracy for hSCP model is $59.3\%$ and for the rshSCP is $33.6\%$. Using the second architecture, the cross-validation accuracy for hSCP model is $58.7\%$ and for the rshSCP is $33.4\%$. This suggests that our model can reduce the prediction capability to predict site.

**Age prediction** We used subject specific information ($\mathbf{\Lambda}$) to predict age of each subject as the metric to check if our method is able to preserve age related biological variation. Table 7 shows 10 fold cross validation mean absolute error (MAE) using random forest . From the table, we can see that the proposed method has comparable performance as the hSCP suggesting that it preserves age related biological variance. More details about the experiment are given in Appendix D.2.

### 3.3 Analysis of components

A robust method should be able to reduce non-biological variability caused by site and scanner while retaining biological variability. In this study, we look at brain aging-related associations and leave analysis with other variables for future work. We also discuss the difference between the components with and reduced site effects. We selected the subjects with age greater than $60$ to find an association between brain aging and the components derived from hSCP and rshSCP. The total number of subjects having an age greater than $60$ is $2746$. We first computed Spearman correlation between subject-specific information ($\mathbf{\Lambda}$) and their respective age and used $0.05$ as the significance level for the hypothesis test of no correlation against the alternative hypothesis of a nonzero correlation. We derived 10 fine-scale ($1 - 10$) and 4 coarse-scale components (I-IV) because of the high split sample reproducibility and easier interpretation of each component. The correlation and $p$-values are displayed in Table 15 and 16 in Appendix D.2 for the hSCP and rshSCP.

We first compare components from the two methods. Figure 1 shows the components derived from hSCP and the proposed method. The first row of the figure displays the components with

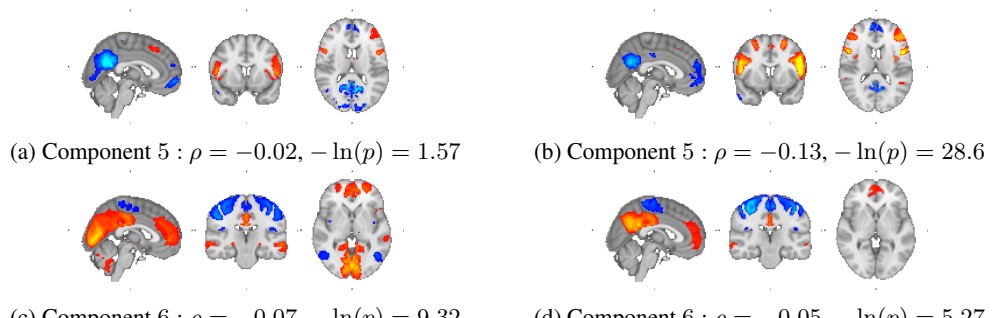

(a) Component 5 : $\rho = -0.02, -\ln(p) = 1.57$     (b) Component 5 : $\rho = -0.13, -\ln(p) = 28.6$

(c) Component 6 : $\rho = -0.07, -\ln(p) = 9.32$     (d) Component 6 : $\rho = -0.05, -\ln(p) = 5.27$

Figure 1: Left column ((a) & (c)) displays the components estimated using hSCP and right column ((b) & (d)) displays the components estimated using rshSCP. Red and blue regions are anti-correlated with each other but are correlated among themselves. The colors are not associated with negative or positive correlation since they can be swapped without affecting the final inference.

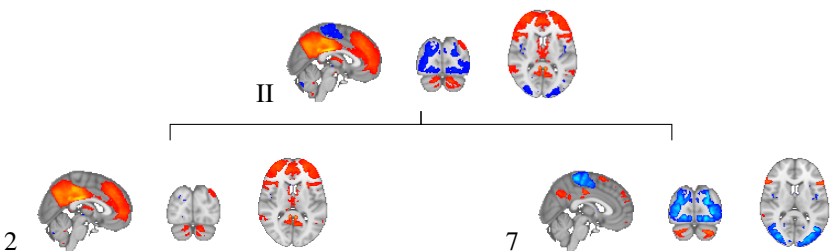

Figure 2: One of the hierarchical components derived from rshSCP comprising of component 2 and 7 at fine scale and component II at coarse scale.

anti-correlation between Default Mode Network (DMN) and Dorsal Attention Network (DAN). The component derived using hSCP has a part of the visual area positively associated with DMN, but the opposite is true, as shown by the previous sparse connectivity patterns [23]. On the other hand, the component with reduced site effects is cleaner since it does not include that relation. This component has a negative correlation with age which has been previously shown in resting-state fMRI and task-based fMRI [38]. The magnitude of anti-correlation has been connected to individual differences in task performances in healthy young adults [39]. However, in the case of older adults, the behavioral implications of reduced anti-correlation remain unclear. The second row of the Figure 1 displays another set of components for comparison. The components stores information about the anti-correlation between DMN and sensorimotor, which aligns with the previous literature [40]. But the addition of a positive correlation of DMN with visual areas will cause misleading inference since it contradicts the previous SCPs and studies. Hence making an inference without removing the site effects can be misleading. Discussion of the association between other relevant components and aging is in Appendix D.2 showing biological interpretability of the components in brain aging. From the results, we can see that there is an increase (or decrease in anti-correlation) in connectivity between different networks in the aging brain. This suggests that there is a reorganization of the aging brain aligning with the previous findings [41]. This can serve as a base to explore rshSCP as a biomarker of neurodegenerative diseases.

Figure 2 displays one of the hierarchical components with coarse-scale component storing relation between different fine-scale components comprising DMN, sensorimotor, and visual areas, previously studied by [40]. These findings give evidence that even after removal site effects, the components can have a meaningful interpretation. The results indicate that our approach can extract robust informative patterns without using traditional seed-based methods that are dependent on the knowledge of the seed region of interest.

# 4 Conclusion

In this work, we have presented a method for estimating site effects in hSCP. We formulated the problem as a minimax non-convex optimization problem and solved it using AMSgrad. We also propose a simple initialization procedure to make the optimization procedure deterministic and improve the performance on an average on a simulated and real dataset. Experimentally, using a simulated dataset, we showed that our method accurately estimates the ground truth compared to the vanilla method with better reproducibility. On the real dataset, we show that the proposed method can capture components with a better split sample and leave one site out reproducibility without losing biological interpretability and information. We also show that without removal of site effects, we can have a noisy estimate of sparse components resulting in misleading downstream analysis.

Below we mention some directions for future research. First, it would be interesting to consider the framework for the analysis of task-induced activity to investigate the extent of site effect and corrections on underlying networks activated by the task. Second, one could look at the changes in the associations of hSCPs with various clinical variables such as Mini-mental score, Digit Span Forward score, etc., after removing site effects. Third, we can also look low-dimensional modeling of $\mathbf{V}$ along with sparse constraints which has been used several robust matrix factorization problems. Since we have only shown age related biological preservation, future studies will focus on whether the proposed method preserves components associated with other demographic, clinical phenotypes, and pathological biomarkers.

There are few weaknesses of our proposed model, which also adds directions for future work. First, our method only captures linear site effects, it would be interesting to see if explicitly capturing non-linear site effects can improve the performance of the model. Second, the result of the optimization algorithm depends on the initialization procedure, which has been shown to perform well on the simulated dataset and real dataset but can be sub-optimal.

# 5 Broader Impact

In this work, we provide a new method to diminish site effects in hSCPs and robustly estimate the components in multi-center studies. The formulation used in the method is not limited to hSCP. It can be easily extended to various matrix factorization approaches such as Independent Component Analysis, Non-negative Matrix Factorization, Dictionary Learning, etc., to improve the reproducibility of functional networks/components. Our work not only has broader applicability in terms of methods used for estimation of components but also to different types of neuroscience data, which includes EEG, MEG, etc.

Meanwhile, it should be noted that any unsupervised machine learning model has risk associated with it in terms of accuracy of the model; our work is no departure from this. It can lead to a negative impact since the reproducibility of the components is not $100\%$, which can lead to misleading clinical and biological interpretability of some of the components. This can be prevented by analyzing the components having high reproducibility as shown in section 3.2 and 3.3.

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
