# A    Related Work: Extended

## A.1    Methods for Tackling Site Effects

One of the first investigations of batch effects in rs-fMRI was performed by Olivetti et al. [42] using extremely randomized trees along with dissimilarity representation. One of the common methods to remove site effects is the harmonization of data. Harmonization of fMRI data especially derived measures, is very nascent, even though it is much needed with the growing number of multi-site data sets [43]. Recently, Yu et al. [8] used ComBat harmonization [26] to remove site effects in connectivity matrices. However, ComBat and its variants such as ComBat-GAM [27] can not be directly applied to connectivity matrices since it can destroy the structure of the connectivity matrix and semi definiteness of the connectivity matrix. A similar difficulty arises when applying ComBat based harmonization to other structured data. Another approach is not to remove site effects, but directly use site information for downstream analysis such as age prediction, finding associations with various clinical variables, etc. Kia et al. [44], Bayer et al. [45] used normative modeling for the age prediction task while keeping the site as one of the predictors. One limitation of the method is that without removing the site effects, the biomarkers can not be used for downstream analysis by the clinician, psychiatrist, etc., directly, which is one of the goals of the hSCP.

Recent work by Vega and Greiner [46] analyzed the impact of covariate analysis, z-score normalization, and whitening on batch effects. Domain adaption has also been introduced in removing batch effects in rs-fMRI data. Domain adaption techniques aim to learn from multiple sources and generalize the model to perform well on a new related target site. Extensive work has been done on unsupervised domain adaptation approaches [47, 48]. Several methods have been introduced for domain adaptation such as Multi-source Domain Adversarial Networks [49], Multi-Domain Matching Networks [50], Moment Matching [51], etc. Readers can refer to the detailed survey by Zhao et al. [52]. In multi-site fMRI data, Wang et al. [53] introduced a low-rank domain to remove batch effects. Other recent approached include transport-based joint distribution alignment [54] and federated learning [55] for fMRI data.

## A.2    Hierarchical Decomposition

The hierarchical organization has been observed in large-scale computer architectures [56], communication systems [57], and social networks [58]. Such an organization provides a unique solution to balancing information within a group at a single scale and between groups at multiple scales. It also promotes optimal and efficient information processing and transmission in real-world information processing systems [59]. The hierarchical organization is also seen in natural information processing systems such as the human brain, where this organization is present both in space [60] and time [61]. Our understanding of hierarchical network structure in the brain is limited, despite having fundamental importance, due to its complex nature.

Several different fields have evolved with a particular method to analyze brain organization. One such widely used method is community detection. Several interesting multi-scale community detection methods have been developed for estimating the underlying hierarchical organization of human brain connectivity [62–65]. The major limitations of the community detection approaches are one or more than one of the following: 1) the assumption of independent components, 2) not capturing heterogeneity in the data, and 3) inability to detect weights while estimating links. In addition, community detection approaches remove negative edge links before the analysis as they treat them repulsion. Whereas, in the fMRI, negative links carry essential that can play an essential role in analyzing neuropsychiatric disorders [66] has a substantial physiological basis [67, 68]. Another set of methods that have recently evolved is based on deep learning approaches [69, 20, 70]. These methods have reported the meaningful hierarchical temporal organization of fMRI time series in the task-evoked fMRI data. Still, there are several disadvantages: 1) non existence of positively and negatively correlated nodes in a component, 2) inability to capture heterogeneity in the data, and 3) "black box" results lacking explainability mainly due to non-linearity in the hierarchical associations.

# B Algorithm

## B.1 Gradient Calculations

In this section, we define gradients used for alternating gradient descent. Let

$$\tilde{\mathbf{W}}_0 = \mathbf{W}_0 = \mathbf{I}_P, \qquad \mathbf{Y}_r = \prod_{j=0}^{r} \mathbf{W}_j, \qquad \tilde{\mathbf{Y}}_r = \prod_{j=0}^{r} \tilde{\mathbf{W}}_j,$$

$$\mathbf{T}_{m,n}^r = (\prod_{j=1}^{m-r} \mathbf{W}_j)\boldsymbol{\Lambda}_{m-r}^n(\prod_{j=1}^{m-r} \mathbf{W}_j)^\top, \qquad \tilde{\mathbf{T}}_{m,n}^r = (\prod_{j=1}^{m-r} \tilde{\mathbf{W}}_j)\boldsymbol{\Lambda}_{m-r}^n(\prod_{j=1}^{m-r} \tilde{\mathbf{W}}_j)^\top,$$

$$\mathbf{X}_r^n = \boldsymbol{\Theta}^n - \mathbf{U}_r^s\mathbf{V}_r, \qquad \mathbf{Z}_r^n = \boldsymbol{\Theta}^n - (\prod_{j=1}^{r} \mathbf{W}_j)\boldsymbol{\Lambda}_r^n(\prod_{n=1}^{r} \mathbf{W}_n)^\top,$$

where $n \in \mathcal{I}_s$, $\mathbf{X}_r^n$ stores the information after removing site effects from $\boldsymbol{\Theta}^n$ and $\mathbf{Z}_r^n$ stores the information after removing subject-wise and shared component information at the $r$th level. We first define gradient for updating adversarial perturbations $\tilde{\mathbf{W}}_\mathbf{r}$. The gradient of classifier loss with respect to $\mathcal{D}$ is calculated using automatic differentiation provided by MATLAB. The objective function is $F = \alpha\|\hat{\mathbf{W}}_r - \mathbf{W}_r\|_F^2 + H(\tilde{\mathcal{W}}, \mathcal{D}, \mathcal{P})$ and gradient with respect to $\tilde{\mathbf{W}}_\mathbf{r}$ will be

$$\frac{F}{\partial \tilde{\mathbf{W}}_\mathbf{r}} = 2\alpha(\hat{\mathbf{W}}_r - \mathbf{W}_r) + \frac{\partial H(\tilde{\mathcal{W}}, \mathcal{D}, \mathcal{C})}{\partial \tilde{\mathbf{W}}_r}$$

$$= 2\alpha(\hat{\mathbf{W}}_r - \mathbf{W}_r) + \sum_{n=1}^{N}\sum_{j=r}^{K}\left(-4\tilde{\mathbf{Y}}_{r-1}^\top\boldsymbol{\Gamma}^n\tilde{\mathbf{Y}}_{r-1}\tilde{\mathbf{W}}_r\tilde{\mathbf{T}}_{j,n}^r\right.$$

$$\left. + 4\tilde{\mathbf{Y}}_{r-1}^\top\tilde{\mathbf{Y}}_{r-1}\tilde{\mathbf{W}}_r\tilde{\mathbf{T}}_{j,n}^r\tilde{\mathbf{W}}_r^\top\tilde{\mathbf{Y}}_{r-1}^\top\tilde{\mathbf{Y}}_{r-1}\tilde{\mathbf{W}}_r\tilde{\mathbf{T}}_{j,n}^r\right).$$

We now define gradients for updating model parameters. The gradient of objective function $J$ with respect to $\boldsymbol{\Lambda}_r^i$ is:

$$\frac{\partial J}{\partial \boldsymbol{\Lambda}_r^i} = \frac{\partial H(\tilde{\mathcal{W}}, \mathcal{D}, \mathcal{C})}{\partial \boldsymbol{\Lambda}_r^i} + \beta\frac{\partial H(\mathcal{W}, \mathcal{D}, \mathcal{C})}{\partial \boldsymbol{\Lambda}_r^i} + \gamma\frac{\partial \mathcal{L}(\zeta, \mathcal{D}, \mathbf{y})}{\partial \boldsymbol{\Lambda}_r^i}$$

$$= \left[(-2\tilde{\mathbf{Y}}_r^\top\mathbf{X}_r^i\tilde{\mathbf{Y}}_r + 2\tilde{\mathbf{Y}}_r^\top\tilde{\mathbf{Y}}_r\boldsymbol{\Lambda}_r^i\tilde{\mathbf{Y}}_r^\top\tilde{\mathbf{Y}}_r) + \beta(-2\mathbf{Y}_r^\top\mathbf{X}_r^i\mathbf{Y}_r + 2\mathbf{Y}_r^\top\mathbf{Y}_r\boldsymbol{\Lambda}_r^i\mathbf{Y}_r^\top\mathbf{Y}_r)\right] \circ \mathbf{I}_{k_r} + \gamma\mathbf{F},$$

where is $\mathbf{F}$ i.e $\frac{\partial \mathcal{L}(\zeta, \mathcal{D}, \mathbf{y})}{\partial \boldsymbol{\Lambda}_r^i}$ is calcualted using automatic differentiation toolbox in MATLAB. The gradient of $J$ with respect to $\mathbf{W}_r$ is:

$$\frac{\partial J}{\partial \mathbf{W}_r} = \frac{\partial H(\mathcal{W}, \mathcal{D}, \mathcal{C})}{\partial \mathbf{W}_r}$$

$$= \sum_{n=1}^{N}\sum_{j=r}^{K}-4\mathbf{Y}_{r-1}^\top\mathbf{X}_n\mathbf{Y}_{r-1}\mathbf{W}_r\mathbf{T}_{j,n}^r + 4\mathbf{Y}_{r-1}^\top\mathbf{Y}_{r-1}\mathbf{W}_r\mathbf{T}_{j,n}^r\mathbf{W}_r^\top\mathbf{Y}_{r-1}^\top\mathbf{Y}_{r-1}\mathbf{W}_r\mathbf{T}_{j,n}^r.$$

The gradient $J$ with respect to $\mathbf{U}^s$ and $\mathbf{V}$ are:

$$\frac{\partial J}{\partial \mathbf{U}^s} = \left(\sum_{n=\mathcal{I}_s}(\mathbf{Z}^n - \mathbf{U}^s\mathbf{V})\mathbf{V}^\top\right) \circ \mathbf{I}_p$$

$$\frac{\partial J}{\partial \mathbf{V}} = \sum_{s=1}^{S}\sum_{n\in\mathcal{I}_s}\mathbf{U}^s(\mathbf{Z}^n - \mathbf{U}^s\mathbf{V})$$

## B.2 Alternating Minimization

Algorithm 1 describes the complete alternating minimization procedure. $\mathcal{W}$ and $\mathcal{D}$ are initialized using svd $-$ initialization algorithm [24][Algorithm 2], and $\mathcal{U}$ and $\mathbf{V}$ according to the equation 10.

---

**Algorithm 1** rshSCP

---

1: **Input:** Data $\mathcal{C}$, number of connectivity patterns $k_1, \ldots, k_K$ and sparsity $\tau_1, \ldots, \tau_K$ at different level, hyperparameters $\alpha$, $\beta$, $\gamma$ and $\mu$.
2: Initialize $\mathcal{W}$ and $\mathcal{D}$ using $\mathrm{svd-initialization}$
3: Initialize $\mathcal{U}$ and $\mathbf{V}$ using equation 10
4: **repeat**
5:    **for** $r = 1$ **to** $K$ **do**
6:       **if** Starting criterion is met **then**
7:          *Update adversarial perturbations*
8:          $\hat{\mathbf{W}}_r \leftarrow \mathrm{descent}(\hat{\mathbf{W}}_r, \alpha)$
9:          $\mathbf{W}_r \leftarrow \mathrm{descent}(\mathbf{W}_r)$
10:       **if** $r == 1$ **then**
11:          $\mathbf{W}_r \leftarrow \mathrm{proj}_1(\mathbf{W}_r, \tau_r)$
12:       **else**
13:          $\mathbf{W}_r \leftarrow \mathrm{proj}_2(\mathbf{W}_r)$
14:       **for** $n = 1, .., N$ **do**
15:          $\mathbf{\Lambda}_r^n \leftarrow \mathrm{descent}(\mathbf{\Lambda}_r^n, \beta, \gamma)$
16:          $\mathbf{\Lambda}_r^n \leftarrow \mathrm{proj}_2(\mathbf{\Lambda}_r^n)$
17:    **for** $s = 1$ **to** $S$ **do**
18:       $\mathbf{U}^s \leftarrow \mathrm{descent}(\mathbf{U}^s)$
19:    $\mathbf{V} \leftarrow \mathrm{descent}(\mathbf{V})$
20:    $\mathbf{V} \leftarrow \mathrm{proj}_3(\mathbf{V}, \mu)$
21: **until** Stopping criterion is reached
22: **Output:** $\mathcal{W}$ and $\mathcal{L}$

---

$\mathrm{proj}_1(\mathbf{W}, \tau)$ operator is used for projecting each column of $\mathbf{W}$ into the intersection of $L_1$ and $L_\infty$ ball [71], $\mathrm{proj}_2$ operator is used for projecting a matrix onto $\mathbb{R}_+$ by zeroing out all then negative values in the matrix, and $\mathrm{proj}_3$ operator is used for projection onto $L_1$ ball. We use AMSGrad [72] denoted as descent in the algorithm with gradients defined in the previous section for performing gradient descent for all the variables. $\beta_1$ and $\beta_2$ are kept to be 0.9 and 0.999 in AMSGrad based on the experiments of [24, 25]. We start adversarial training only after the convergence of all the variables. We found that the algorithm uses 200 iterations to reach convergence initially, as shown in Figure 3. The reason being that adversarial learning can start from an optimal point on which it can improve upon if there is overfitting. When the adversarial learning starts, first, the adversarial perturbations are generated by performing gradient descent on $\hat{\mathbf{W}}_r$, and then the model parameters are updated using gradient descent. This process is repeated until the convergence criteria is met.

### B.3 Convergence results

We empirically validate the convergence of Algorithm 1 using the reconstruction error:

$$\frac{\sum_{s=1}^S \sum_{n \in \mathcal{I}_s} \sum_{r=1}^K ||\mathbf{\Theta}^i - (\prod_{j=1}^r \mathbf{W}_j)\mathbf{\Lambda}_r^i(\prod_{j=1}^r \mathbf{W}_j)^\top - \mathbf{U}^s\mathbf{V}||_F^2}{\sum_{n=1}^N \sum_{r=1}^K ||\mathbf{\Theta}^i||_F^2}.$$

Figure 3 shows the convergence of the algorithm on the complete dataset.

## C  Data Preprocessing

The pooled dataset included scans of participants with absence of any known diagnosis of a neurological or psychiatric disorder. FMRIB Software [73] is used for initial pre-processing as a part of the UK Biobank pipeline. The steps included the removal of the first five volumes, head movement correction using FSL's MCFLIRT [73], global 4D mean intensity normalization, and temporal high-pass filtering ($> 0.01$ Hz).

After standard pre-processing steps, we applied FIX (FMRIB's ICA-based Xnoiseifier) [74, 75] to remove structured artefacts. In the next step, functional images were co-registered to T1 using FLIRT

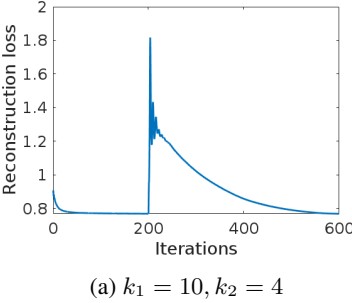

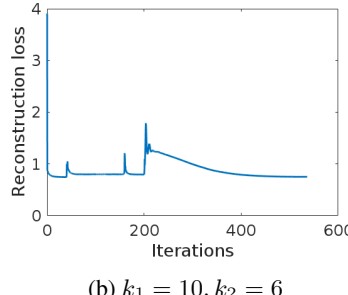

(a) $k_1 = 10, k_2 = 4$            (b) $k_1 = 10, k_2 = 6$

Figure 3: Convergence of rshSCP algorithm using the complete dataset for different values of $k_2$. In the figure, for the first 200 iterations, the algorithm converges without the adversarial perturbations. As the adversarial perturbations are introduced, the loss starts to oscillate where the adversarial perturbations force the algorithm to deviate from the optimal value. In defense, we minimize the objective function until convergence is reached.

Table 8: 5 fold cross validation accuracy (%) on simulated dataset at one level.

| Method | $k_1 = 8$ | $k_1 = 10$ | $k_1 = 12$ | $k_1 = 14$ |
|---|---|---|---|---|
| hSCP | $97.3 \pm 0.3$ | $98.1 \pm 0.4$ | $97.1 \pm 0.3$ | $97.9 \pm 0.2$ |
| Adv. rshSCP | $65.5 \pm 0.6$ | $67.2 \pm 0.5$ | $67.5 \pm 0.5$ | $68.1 \pm 0.7$ |

with BBR as the cost function, and T1-weighted images were registered to the MNI152 template using FSL's FNIRT (non-linear registration). We projected the data into a lower-dimensional space by extracting a set of group Independent Components [76] having dimension 100 from individual subjects. These ICA maps can be considered "parcellations" but contain a continuous range of values and not binary masks. For a given IC map, the group IC spatial maps were mapped onto each subject's resting fMRI time series to derive one representative time series per IC component using Group Information Guided ICA(GIGICA) [77].

Quality control of the dataset is based on below metrics-

1. Mean Relative (frame-wise) Displacement (MRD): We used MRD calculated by MCFLIRT to quantify head motion [78]. We set a threshold 0.2mm.

2. Time course Signal to Noise Ratio (tSNR): tSNR is an important metric for evaluating the ability of the fMRI acquisition to detect neural signal changes in the time series. It is is defined as the ratio of mean intensity and standard deviation across time within the evaluated Region of Interest [79]. We excluded the subjects having temporal SNR less than 100.

3. Framewise Displacement (FD): It evaluates the head motion of each volume compared to the previous volume [80, 81]. We set the threshold for FD to be 0.2 mm [80, 82].

## D   Additional experiments

### D.1   Simualted Dataset

**Site prediction.** To check if the estimated subject information ($\Lambda$) has reduced predictive power to predict the site to which the subject belonged, we performed a 5 fold cross-validation using SVM with RBF kernel. For prediction using SVM, we directly use the inbuilt function of MATLAB. Since we already used feed forward network, we decided to use a different model. Prediction results for site using $\Lambda$ are displayed in Table 8. We can see that there is an average drop of $20\%$ accuracy.

**Reproducibility.** Table 9 and 10 shows the split sample reproducibility of different methods on two level simulated dataset.

Table 9: Reproducbility on simulated dataset ($k_2 = 4$).

| Method | $k_1 = 8$ | $k_1 = 10$ | $k_1 = 12$ | $k_1 = 14$ |
|---|---|---|---|---|
| hSCP | $0.801 \pm 0.037$ | $0.805 \pm 0.042$ | $0.787 \pm 0.041$ | $0.772 \pm 0.037$ |
| ComBat hSCP | $0.776 \pm 0.041$ | $0.756 \pm 0.044$ | $0.753 \pm 0.045$ | $0.745 \pm 0.038$ |
| Adv. hSCP | $0.808 \pm 0.036$ | $0.824 \pm 0.034$ | $0.799 \pm 0.030$ | $0.783 \pm 0.038$ |
| rshSCP | $0.850 \pm 0.037$ | $0.853 \pm 0.031$ | $0.839 \pm 0.035$ | $0.805 \pm 0.034$ |
| Adv. rshSCP | $\mathbf{0.852 \pm 0.036}$ | $\mathbf{0.861 \pm 0.038}$ | $\mathbf{0.842 \pm 0.043}$ | $\mathbf{0.813 \pm 0.035}$ |

Table 10: Reproducbility on simulated dataset ($k_2 = 6$).

| Method | $k_1 = 8$ | $k_1 = 10$ | $k_1 = 12$ | $k_1 = 14$ |
|---|---|---|---|---|
| hSCP | $0.786 \pm 0.041$ | $0.801 \pm 0.039$ | $0.771 \pm 0.042$ | $0.769 \pm 0.038$ |
| ComBat hSCP | $0.779 \pm 0.044$ | $0.770 \pm 0.041$ | $0.742 \pm 0.040$ | $0.734 \pm 0.045$ |
| Adv. hSCP | $0.793 \pm 0.036$ | $0.828 \pm 0.038$ | $0.789 \pm 0.035$ | $0.762 \pm 0.036$ |
| rshSCP | $0.833 \pm 0.038$ | $0.846 \pm 0.031$ | $0.831 \pm 0.032$ | $0.795 \pm 0.034$ |
| Adv. rshSCP | $\mathbf{0.841 \pm 0.039}$ | $\mathbf{0.851 \pm 0.035}$ | $\mathbf{0.835 \pm 0.033}$ | $\mathbf{0.808 \pm 0.039}$ |

Table 11: Change in accuracy of rshSCP with sparsity parameter ($\mu$) of $\mathbf{V}$ on real dataset at one level.

| $\mu$ | $k_1 = 8$ | $k_1 = 10$ | $k_1 = 12$ | $k_1 = 14$ |
|---|---|---|---|---|
| 0.1 | 0.799 | 0.756 | 0.761 | 0.782 |
| 0.5 | 0.873 | 0.865 | 0.843 | 0.867 |
| 1 | 0.801 | 0.789 | 0.767 | 0.754 |

Table 12: Mean absolute error ($k_2 = 6$)

| Method | $k_1 = 10$ | $k_1 = 15$ | $k_1 = 20$ | $k_1 = 25$ |
|---|---|---|---|---|
| hSCP | $6.472 \pm 1.417$ | $6.440 \pm 1.470$ | $6.418 \pm 1.484$ | $6.401 \pm 1.478$ |
| Adv. rshSCP | $6.475 \pm 1.250$ | $6.439 \pm 1.411$ | $6.421 \pm 1.454$ | $6.403 \pm 1.474$ |

Table 13: Split-sample reproducbility on real dataset ($k_2 = 6$).

| Method | $k_1 = 10$ | $k_1 = 15$ | $k_1 = 20$ | $k_1 = 25$ |
|---|---|---|---|---|
| hSCP | $0.691 \pm 0.034$ | $0.688 \pm 0.034$ | $0.677 \pm 0.029$ | $0.668 \pm 0.032$ |
| ComBat hSCP | $0.670 \pm 0.026$ | $0.664 \pm 0.028$ | $0.635 \pm 0.030$ | $0.626 \pm 0.028$ |
| Adv. hSCP | $0.701 \pm 0.026$ | $0.696 \pm 0.029$ | $0.681 \pm 0.028$ | $0.679 \pm 0.031$ |
| rshSCP | $0.776 \pm 0.027$ | $0.748 \pm 0.029$ | $0.722 \pm 0.032$ | $0.721 \pm 0.024$ |
| Adv. rshSCP | $\mathbf{0.779 \pm 0.029}$ | $\mathbf{0.751 \pm 0.026}$ | $\mathbf{0.731 \pm 0.027}$ | $\mathbf{0.732 \pm 0.025}$ |

### D.2 Real Dataset

**Reproducibility.** Table 13 and Table 14 shows split sample reproducibility and leave one site out reproducibility respectively at two level for $k_2 = 6$ and multiple values of $k_1$.

**Age prediction.** We used subject specific information ($\mathbf{\Lambda}$) having total $k_1 + k_2$ features from the two layers to predict age of each subject. We used Bootstrap-aggregated (bagged) decision trees to perform regression with 400 trees for each site separately. Table 12 shows the average and standard deviation of 10 fold cross validation mean absolute error (MAE) across site for varied values of $k_1$ and $k_2 = 6$. We decided to perform age prediction of each site separately because the age is confounded by the site. The correlation between age and site is $0.24$ and reduction in site effects would reduce the prediction capability in the pooled setting.

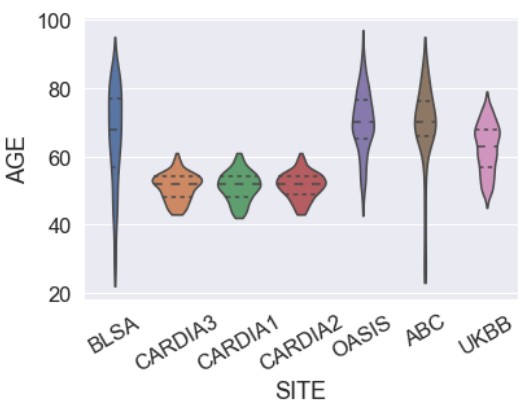

Figure 4: Violin plot of age for different sites.

Table 14: Leave one site out reproducbility on real dataset($k_2 = 6$).

| Method | $k_1 = 10$ | $k_1 = 15$ | $k_1 = 20$ | $k_1 = 25$ |
|---|---|---|---|---|
| hSCP | $0.637 \pm 0.035$ | $0.600 \pm 0.036$ | $0.578 \pm 0.031$ | $0.560 \pm 0.034$ |
| ComBat hSCP | $0.618 \pm 0.037$ | $0.589 \pm 0.033$ | $0.543 \pm 0.035$ | $0.521 \pm 0.032$ |
| Adv. hSCP | $0.642 \pm 0.028$ | $0.608 \pm 0.021$ | $0.585 \pm 0.023$ | $0.572 \pm 0.019$ |
| rshSCP | $0.701 \pm 0.032$ | $0.691 \pm 0.034$ | $0.668 \pm 0.031$ | $0.659 \pm 0.029$ |
| Adv. rshSCP | $\mathbf{0.703 \pm 0.033}$ | $\mathbf{0.695 \pm 0.035}$ | $\mathbf{0.672 \pm 0.030}$ | $\mathbf{0.666 \pm 0.031}$ |

Table 15: Spearman correlation ($\rho$) and p-value of age ($> 60$) with $\mathbf{\Lambda}_1$ computed from hSCP (a) and Adv. rshSCP (b).

| | | 1 | 2 | 3 | 4 | 5 | 6 | 7 | 8 | 9 | 10 |
|---|---|---|---|---|---|---|---|---|---|---|---|
| a | $\rho$ | 0.07 | 0.0 | 0.09 | $-0.12$ | $-0.02$ | $-0.07$ | $-0.15$ | $-0.04$ | $-0.06$ | 0.02 |
| | $-\ln(p)$ | 9.28 | 0.27 | 15.0 | 35.3 | 1.57 | 9.32 | 33.3 | 3.10 | 5.98 | 1.27 |
| b | $\rho$ | 0.05 | $-0.02$ | 0.11 | $-0.07$ | $-0.13$ | $-0.05$ | $-0.03$ | 0.0 | $-0.02$ | 0.02 |
| | $-\ln(p)$ | 5.29 | 0.96 | 20.0 | 8.77 | 28.6 | 5.27 | 2.46 | 0.16 | 1.67 | 1.44 |

Table 16: Spearman correlation ($\rho$) and p-value of age ($> 60$) with $\mathbf{\Lambda}_2$ computed from hSCP and Adv. rshSCP.

| | | I | II | III | IV |
|---|---|---|---|---|---|
| hSCP | $\rho$ | 0.02 | $-0.04$ | $-0.08$ | $-0.13$ |
| | $-\ln(p)$ | 1.69 | 3.68 | 9.83 | 24.7 |
| Adv. rshSCP | $\rho$ | 0.05 | $-0.05$ | $-0.06$ | $-0.09$ |
| | $-\ln(p)$ | 4.95 | 5.34 | 6.98 | 13.8 |

**Association with brain aging.** We computed spearman correlation of age ($> 60$) with $\mathbf{\Lambda}_1$ and $\mathbf{\Lambda}_2$. We then calculated p-values for the hypothesis test of no correlation against the alternative hypothesis of a nonzero correlation and are converted to $-\ln(p)$, where ln is log base 2. Table 15 and 16 displays spearman correlation and negative log p-value. The total number of subjects with age greater than 60 is 2746. In the case of negative log base 2, if the value is greater than 2.99 then we consider it statistically significant, equivalent to p-value less than 0.05. Figure 5 shows anti-correlation between DMN, and Salience Network (SN) and Central Executive Network (CEN), and negative correlation with age. Previously, Nagel et al. [83] has found a decrease in the functional coupling between DMN and the premotor cortex, corroborating our results.

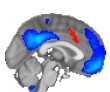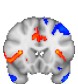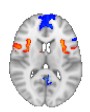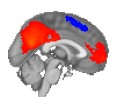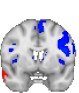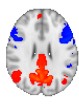

    (a) Component 7 : $\rho = -0.03, -\ln(p) = 2.46$      (b) Component 4 : $\rho = -0.07, -\ln(p) = 8.77$

Figure 5: (a) Anticorrelation between Default Mode Network (DMN) and Salience Network (SN) and (b) Anticorrelation between Default Mode Network (DMN) and Central Executive Network (CEN). Red and blue regions are anti-correlated with each other but are correlated among themselves. The colors are not associated with negative or positive correlation since they can be swapped without affecting the final inference.

## E   Between-network connectivity in aging

In this section, we discuss related work on changes in between-network connectivity in older adults. Geerligs et al. [84] published one of the earliest studies on changes in between-network connectivity in older adults using seed-based analysis while participants performed an oddball task. They observed stronger connectivity (or weaker anticorrelations) between distinct functional networks. For example, they found age-related connectivity increases between the DMN, and the somatosensory and the CEN, which aligns with the results of current work. Several other studies reported similar results using different approaches [85, 86]. The DAN and DMN appear to show strong anticorrelations due to their presence in externally directed and internally directed cognition. Spreng et al. [38] used both resting and task data to show a decrease in anticorrelations between these networks in older compared to younger.

The increase in connectivity between different networks can be thought of as a decrease in the segregation of networks. Previous studies have indicated that this decrease in segregation causes a reduction in the specialization of specialized networks, affecting information processing of the human brain [87]. Grady et al. [88] analyzed the connections between DMN, DAN and CEN networks and observed a lower index of segregation in older as compared to young. Our results also indicate a decrease in anticorrelation between various networks, which can be thought of decrease in the segregation of networks, resulting in reorganization of the human brain in old age.