# OpenReview forum: "Learning Robust Hierarchical Patterns of Human Brain across Many fMRI Studies"
_NeurIPS.cc/2021/Conference — NeurIPS 2021 Poster_

### Official Review · Reviewer_qpQL · 2021-07-11

**Rating:** 7
**Confidence:** 3

**Summary:**

This paper proposes a method called rshSCP (robust-to-site hSCP) to estimate hierarchical sparse connectivity patterns (hSCPs) from multi-site fMRI data while reducing the impact of site-specific variation due to scanner and experiment details rather than biological effects. Their approach is applied on top of the previous hSCP method as well as an extension that uses adversarial learning to improve reproducibility of estimated patterns (Adv. hSCP). The proposed method includes an additive component to the decomposition of the correlation matrix in hSCP, which models site effects as a matrix factorization with diagonal site-specific and full shared (across-site) matrix components. The method then additionally applies an adversarial classifier (a two hidden-layer feed-forward neural network) to predict the site from the subject-specific components of the hSCP decomposition. The entire optimization is formulated as a minimax problem, where the classifier maximizes its ability to predict site information while the remaining parameters are optimized to reconstruct the correlation matrices while minimizing the classifier's accuracy. The authors also propose an SVD-based initialization which improves reproducibility of the estimated components. Experiments on simulated data as well as fMRI data from multi-center imaging studies show that rshSCP improves accuracy (on simulated data) as well as reproducibility (on simulated and real data) of the estimated components, both when applied with hSCP and Adv. hSCP. The proposed method also leads to a reduction in accuracy when predicting site from the estimated subject-specific components using an SVM, as desired. Predicting age from the estimated subject-specific components with a random forest and visualizing the shared components shows that rshSCP preserves relevant biologically-related variation.

**Ethical Concerns:**

No ethical concerns come to mind.

**Limitations And Societal Impact:**

Yes, the authors discuss the potential negative impact and limitations of their work when considering the validity/interpretability of findings in neuroscience.

**Main Review:**

Overall, the paper is clear and well-written. The primary contribution is the inclusion of an additional additive term to the hSCP decomposition, as well as an adversarial loss and classifier to remove site-specific variation from the learned components. While an adversarial approach to removing confounding effects is not novel, this seems to be its first application to hSCP and the authors carefully demonstrate the ability of this technique to learn more accurate and reproducible components than versions of hSCP that do not use this technique.

One interesting comparison that could have been made is applying a harmonization method (e.g., ComBat) to ${\bf \Lambda}$ directly (as the authors discuss), then addressing the issue of altered relative contributions of the global components by fixing the harmonized ${\bf \Lambda}$ values and re-fitting the ${\bf W}$ matrices. Determining whether this approach achieves comparable reproducibility and/or interpretability of the learned components would provide a simple baseline to compare to the adversarial approach. However, the authors demonstrate the utility of their proposed method rigorously enough even when compared to other versions of hSCP that I don't feel the lack of this comparison to be a major issue.

Another choice that could be altered is using an SVM to demonstrate the reduced ability to predict site information from the learned subject-specific components. The authors do not justify this other than to say "we already used feed forward network". It is unclear whether a feed forward neural network (such as the architecture used to train the adversarial classifier) would be better at extracting site information even from the learned rshSCP components, and showing the results with the same or similar choice of classifier would strengthen that result.

Other minor comments:
- The subscript for the second ${\bf W}$ in the top row of Equation 1 should be $j$, not $n$.
- If my understanding is correct, the cross-entropy loss term $\mathcal{L}$ in Equation 7 should have a negative sign (so that the cross-entropy is maximized with respect to the hSCP parameters and minimized with respect to $\zeta$ in the joint optimization)

**Time Spent Reviewing:**

3

---

> ### Author Response · Authors · 2021-08-11
> **Reply to Reviewer qpQL**
>
> We thank the reviewer for their detailed comments and for suggestions for improving the paper. We address the comments below:
>
> 1. **One interesting comparison that could have been made is applying a harmonization method (e.g., ComBat) to Lambda directly (as the authors discuss), then addressing the issue of altered relative contributions of the global components by fixing the harmonized Lambda values and re-fitting the W matrices. Determining whether this approach achieves comparable reproducibility and/or interpretability of the learned components would provide a simple baseline to compare to the adversarial approach. However, the authors demonstrate the utility of their proposed method rigorously enough even when compared to other versions of hSCP that I don't feel the lack of this comparison to be a major issue.**
>
> We thank the reviewer for the interesting suggestion. We implemented a baseline as suggested by you.We first ran hSCP and used ComBat on the extracted $\mathbf{\Lambda}^n$ to get harmonized subject specific information $\mathbf{\Delta}^n$ for each subject. We then re-fitted $\mathbf{W}_1$ and $\mathbf{W}_2$ matrices using the below decomposition-
>
> $$\mathbf{\Theta}^n \approx \mathbf{W}_1 (\mathbf{\Delta}^n_1+\mathbf{S}_1) \mathbf{W}_1^\top \approx \mathbf{W}_1 \mathbf{W}_2(\mathbf{\Delta}^n_2+\mathbf{S}_2) (\mathbf{W}_1\mathbf{W}_2)^\top $$
>
> We added a diagonal shift matrix $\mathbf{S}$ such that $\mathbf{\Delta}^n+\mathbf{S}$ is positive for each subject and performed the optimization to estimate $\mathbf{W}$ and $\mathbf{S}$. Below is the split sample reproducibility on real dataset for $k_2 = 4$ and varied values of $k_1$
>
> $$k_1 = 10  :  0.673\pm0.049 \quad k_1 = 10 :   0.641\pm0.051  \quad k_1 = 20 : 0.639\pm0.031 \quad k_1 = 25 :  0.611\pm0.038$$,
>
> and leave one site out reproducibility on the real dataset for $k_2 = 4$ and varied values of $k_1$
>
> $$k_1 = 10  : 0.614\pm0.042 \quad k_1 = 15  :  0.594\pm0.035  \quad  k_1 = 20 :  0.542\pm0.041 \quad k_1 = 125 :  0.528\pm0.039$$
>
> From the above results, we can see that this baseline performs worse than vanilla hSCP. One reason for this might be that the harmonized lambda extracted using ComBat might not necessarily result in optimal $\mathbf{W}$, which is highly reproducible. This result bolsters our method that we need a joint optimization procedure to obtain W and lambda with reduced site effects. We have similar results for the simulated dataset as well. We update our manuscript with these experiments.
>
> 2. **Another choice that could be altered is using an SVM to demonstrate the reduced ability to predict site information from the learned subject-specific components. The authors do not justify this other than to say "we already used feed forward network". It is unclear whether a feed forward neural network (such as the architecture used to train the adversarial classifier) would be better at extracting site information even from the learned rshSCP components, and showing the results with the same or similar choice of classifier would strengthen that result.**
>
> This is a great point. We ran our experiment using two different feed forward networks with the following architectures:
>
> (a) A fully connected layer with $50$ hidden units, dropout layer with the rate $0.2$, ReLU, a fully-connected layer with $4$ hidden units and a softmax layer.\
> (b) A fully connected layer with $50$ hidden units, dropout layer with the rate $0.2$, ReLU, a fully-connected layer with $20$ hidden units, dropout layer with rate $0.2$, ReLU, a fully-connected layer with $4$ hidden units and a softmax layer.
>
> Using the first architecture, the cross-validation accuracy for hSCP model is 59.3% and for the rshSCP is 33.6%. Using the second architecture, the cross-validation accuracy for hSCP model is 58.7% and for the rshSCP is 33.4%. We obtain reduced classification accuracy even using the same feed-forward network and slightly modified version of the architecture used to train the adversarial classifier.
>
> 3. **Minor comments:**
>
> Thanks for pointing out the typos. Yes the subscript of W should be j and the cross-entropy term should have a negative sign. We will make the corrections in the final draft.

---

> > ### Comment · Reviewer_qpQL · 2021-09-01
> > **Comment on author response**
> >
> > Thank you for your response, and for the additional clarifying experimental results. I believe this response addresses my concerns, and I am happy to maintain my rating.

---

### Official Review · Reviewer_jmAd · 2021-07-14

**Rating:** 6
**Confidence:** 4

**Summary:**

This paper deals with the important problem of how to pool fMRI data from multiple sites for connectivity estimation. The authors propose a simple extension to adversarial hierarchical sparse connectivity pattern (hSCP) estimation to remove site effects. The argument against adopting methods, such as the widely-used COMBAT, is to preserve the manifold structure of connectivity matrices. The method is tested on simulated data with higher accuracy shown. The method is also tested on real data with higher reproducibility demonstrated and age effects retained.

**Ethical Concerns:**

There are no ethical issues with this paper.

**Limitations And Societal Impact:**

Limitations are briefly mentioned in the conclusions, and for the current prediction task, there is no negative societal impact.

**Main Review:**

Originality
The proposed methods, rshSCP and Adv. rshSCP, build upon previous methods, hSCP and its adversarial extension (Adv. hSCP), partly by using the usual trick of adding a loss to discourage prediction of confounds, in this case site effects. The idea at a high level is not particularly original, but the implementation details are useful. A bit surprised domain adaptation and optimal transport are not cited and discussed.

Quality
The proposed methods are technically sound, but there are a few concerns regarding assumptions and results.
1. Is retaining the positive semidefinite property essential for downstream analysis? Comparisons with DL methods that do not retain positive semidefinite property are needed.
2. Following the theme of modeling the manifold property of correlation matrices, why use l2 loss in (1) as opposed to Gaussian likelihood as in standard Gaussian graphical models or intrinsic distance between covariance?
3. Will there be an identifiability problem since W1*W2 = W1*A* A’*W2 for A orthogonal?
4. For estimating rshSCP with only one site s in the reproducibility experiments, would using the same V estimated from all sites except s for both reproducibility assessment splits result in inflated reproducibility values?
5. How much did site predictability drop using adv. rshSCP compared to adv. hSCP?
6. Age-related biological variability seems to be retained but just on par with hSCP. Does that imply there is no need to account for site effects? What happens if the two studies with quite different age range were removed? Also, how is the performance of adv. hSCP? Overall, to claim biological variability is retained, need to show results for more than just age.
7. Not sure what to make of the qualitative analysis in Figures 1 and 2. Are there no exceptions where adv rshSCP pools “wrong” regions into a network at any of the hierarchical levels?

Clarity
The paper is clearly written but there are some debatable statements in the Introduction.
1. Is hierarchical representation always better than single level parcellation or is it application dependent?
2. Is “binarized” components always better than their continuous counterparts, e.g. derived from PCA and ICA? The argument was higher interpretability with binarization, yet in an earlier statement, it was mentioned that ICA, NMF, and dictionary learning provide “interpretable” components. Along this line, allowing for components to overlap matches biology but this is provided by PCA and ICA.
3. Which biological predictors are likely to be correlated with site effects? If we remove site effects, wouldn’t biological effects be removed as well.

Significance
The problem being tackled is important and the approach in the proposed methods is applicable to other methods though implementation details would likely need to be altered to a certain extent. The results, however, are not particularly impressive. Whether the proposed methods are suitable for EEG and MEG is slightly questionable since temporal dynamics is more important for those modalities.

Summary after Author Responses
While some comments were well addressed, a number of issues remained, such as reproducibility using the same V, how much site predictability dropped using adv. rshSCP compared to adv. hSCP not reported for real data, need for prediction beyond age (which is only correlated with site at 0.2), and using limited subjective prior neuroscience findings to evaluate the extracted components (in a subjective manner). Hence, I will keep my original score.

**Time Spent Reviewing:**

6

---

> ### Author Response · Authors · 2021-08-11
> **Response to Reviewer jmAd**
>
> We thank the reviewer for their detailed comments and observations. We address the comments below:
>
> 1. **A bit surprised domain adaptation and optimal transport are not cited and discussed.**
>
> We thank the reviewer for raising this concern. We will be adding a brief review of domain adaption and optimal transport citing below papers-
>
> (a) Zhao, Sicheng, et al. "Multi-source domain adaptation in the deep learning era: A systematic survey." arXiv preprint arXiv:2002.12169 (2020). (Survey paper)\
> (b) Xu, Ruijia, et al. "Deep cocktail network: Multi-source unsupervised domain adaptation with category shift." Proceedings of the IEEE Conference on Computer Vision and Pattern Recognition. 2018.\
> (c) Redko, Ievgen, et al. "Optimal transport for multi-source domain adaptation under target shift." The 22nd International Conference on Artificial Intelligence and Statistics. PMLR, 2019,
>
> and additional relevant papers.
>
> 2. **Positive semidefinite property essential?**
>
> The idea behind the hSCP model is to find sparse interpretable shared components and their subject-specific weights that need to be non-negative to interpret these weights. Non-negative weights are easier to interpret because a zero weight implies that the subject does not have a component present, and a positive weight will imply the component's contribution.
>
> 3. **Why use l2 loss?**
>
> The correlation matrix in the context of fMRI data is a functional connectivity measure that describes the relationship between different nodes of the human brain, and each element carries equally important information about the relations. Our goal is to decompose the correlation matrix into a non-negative linear combination of rank-one matrices providing us with interpretability, and we want this approximation to be as close as possible to the correlation matrix. A natural choice for this type of estimation where equal weight is given to each element is l2 loss. We do not see any drawback to using l2 distance instead of Log-Euclidean, Riemannian, or Root Euclidean. Further, each element of the correlation matrix is given equal weight in the optimization, and changing the distance metric will change this weightage and will lose interpretability.
>
> 4. **Identifiability problem**
>
> There should not be any identifiability issues since the sparsity constraints, infinity, and trace norm constraints prevent the model from having multiple solutions.
>
> 5. **Inflated reproducibility values?**
>
> The goal of leave one site out reproducibility experiment is to measure the reproducibility between components from an existing dataset and components extracted from a "new" dataset and compare with and without reducing site effects. If a new dataset comes and we would like to reduce the site effects, then we would be using a "known" V to train the model with the assumption that V already has site and scanner information from the previously trained model. The hope is that with the reduction in site effects, the extracted components would be less noisy and match the extracted from the larger dataset, which is shown by our results.
>
> 6. **How much did site predictability drop using adv. rshSCP compared to adv. hSCP?**
>
> Site predictability of adv. hSCP is 92.9% compared to 67% of adv rshSCP under the same settings as mentioned in the paper.
>
> 7. **Biological variability:**
>
> (a) In the paper, we are trying to reduce site variability effects from the components and subject-specific information. If we do not consider site effects, then the extracted components can be noisy (Figure 2), leading to incorrect inferences, as discussed in section 3.3. So we want to obtain clean components which are interpretable while retaining their age-related variability. \
> (b) Can you please elaborate on the question? Did you mean that what happens if we remove studies with quite different age ranges in our current dataset? \
> (c) The components from adv. hSCP were a little noisy, a little better than hSCP, but not comparable to adv rshSCP, hence we decided to focus on the current method.\
> (d) We will edit our manuscript to make it clear that our focus is on the claim that the method preserves age-related variance. We do not wish to claim that it preserves all biological variation since we do not have information on variables associated with the biological variation.
>
> 8. **Qualitative analysis in Figures 1 and 2**
>
> Figure 1 displays two components extracted with and without reduction in site effects. With the corrected version, we get some well-known connections previously studied in the literature using various analyses. But without correction, we get a noisy version of these connections. Figure 2 shows that the hSCP is able to capture "functional polymorphism" where similar components are present at the fine level to capture the functional heterogeneity in the data. They combine to form coarse-level components, capturing heterogeneity at a different scale that fine-level components cannot capture. Our analysis shows that the adv rshSCP is able to extract components that align with literature and do not give any noisy components.
>
> 9. **Is hierarchical representation always better than single level parcellation or is it application dependent?**
>
> We think the type of components/network extraction is application-dependent. Because there is little ground truth on what constitutes a brain network, many people have applied similar decomposition models using ICA and related approaches to extract components/networks suiting their applications. Therefore, we consider hSCP as a method that simply produces components of brain regions that statistically appear to have relatively more synchronous activation patterns and hence could be presumed to be parts of underlying brain networks. hSCP is also a hierarchical dimensionality reduction method possessing certain desirable properties such as sparse, overlapping components, etc., which allows us to investigate hierarchically organized co-activated brain regions and probe heterogeneity of brain activity patterns across individuals. In that case, the optimal or most interpretable solution might be obtained at one scale rather than another, which is another added advantage of hSCP over SCP. In addition, hSCP captures "functional polymorphism" where similar components are present at the fine level to capture the functional heterogeneity in the data. They combine to form coarse-level components, capturing heterogeneity at a different scale that fine-level components cannot capture.
>
> 10. **Is “binarized” components always better than their continuous counterparts, e.g. derived from PCA and ICA?**
>
> Standard approaches such as ICA, NMF, etc., capture interpretable functional brain networks. However, they fail to capture between-network interactions, which aligns with the notion that brain systems often act in concert during complex cognitive functioning and is shown to capture the underlying structure better [1,2]. Figure 2, represents one such component comprising anti-correlation between Default Mode Network and Dorsal Attention Network. Another significant disadvantage of these approaches is their inability to directly quantify inter-subject variability in functional connectivity, requiring additional post-processing analysis.
>
> 11. **Which biological predictors are likely to be correlated with site effects?**
>
> Actually, this was one reason for performing an experiment to check if our method affects age-related variability. Age, sex, cognitive variables such as scores from Trail Making Test, Digit Symbol Substitution Test, etc.,  are some of the biological predictors that are likely to correlate with the site and hence site effects. We do not believe that reduction in site effects would remove biological effects in our method based on our experiment. In our case, the correlation between age and site is 0.23, and our method can still retain age-related effects. We can not experimentally claim that all biological associations are preserved since we don't have data about all known biological predictors. This can be one of the directions for future research, where the goal would be to check if there is a loss in variability in the all-known biological predictors in our pooled dataset.
>
> 12. **Significance:**
>
> We agree that temporal dynamics is more important in EEG and MEG, but recently researchers have been using connectivity matrices or similar matrices for their analysis [3,4]. It would be interesting to see the results using our method (with few modifications).
> We believe that our results are compelling, reason being-
>
> (a) We show that we can have significantly less reproducible and noisy components in a real dataset in multi-site studies without reducing the site effects. Using a simulated dataset, we show that they can be less accurate and reproducible if the site effects are not taken into account. \
> (b) These noisy components, if used for downstream analysis such as finding an association between the components and age or clinical variables, can have misleading results. For example, in our study, we show that these noisy components have a significant correlation with the aging brain, but the end inference will be misleading. Whereas we have results that align with the previous studies when we take site effects into account.\
>
> [1] Andersen, Michael, et al. "Bayesian structure learning for dynamic brain connectivity." International Conference on Artificial Intelligence and Statistics. PMLR, 2018.\
> [2] Eavani, Harini, et al. "Identifying sparse connectivity patterns in the brain using resting-state fMRI." Neuroimage 105 (2015): 286-299.\
> [3] O’Reilly, Christian, et al. "Is functional brain connectivity atypical in autism? A systematic review of EEG and MEG studies." PloS one 12.5 (2017): e0175870.\
> [4] Demuru, M., et al. "Functional and effective whole brain connectivity using magnetoencephalography to identify monozygotic twin pairs." Scientific reports 7.1 (2017): 1-11.

---

### Official Review · Reviewer_8hgE · 2021-07-19

**Rating:** 6
**Confidence:** 4

**Summary:**

This paper introduces the method site-robust hierarchical sparse connectivity patterns (rshscp) and its adversarial extension. It builds on sparse connectivity patterns and its adversarial extension introduced in references 22 and 23. These propose a factorization of correlation matrices extracted for functional connectivity analysis. The factorization consists of a symmetric product of  sparse factors around a central diagonal weighting matrix. The present contribution introduces a site-dependent component into the optimization problem and adds a classifier to predict site, which is then adversarially optimized to make the learned structure lose information about site. In synthetic and real-data experiments it is shown that the method is indeed capable of discarding site information while retaining other relevant information.




**Limitations And Societal Impact:**

There is unlikely to be any societal impact. The possible societal impacts would be positive for medical diagnosis, but fMRI has not established itself as a tool of choice in any field of medical diagnosis.

The obvious limitations of any approach related to fMRI "connectivity" (first and foremost the neglect of first-order driving effects, second the acknowledgement of all possible other confounding effects) should be required to be mentioned in every paper that addresses it. Sadly, this is at odds with the incentives for publishing, so it would need to become a requirement.

**Main Review:**

While this contribution shows that site information can be discounted, I see several issues with it as it stands, which transcend through all evaluation criteria of interest.

Technical questions and comments:
- What is the trace(Lambda) = 1 constraint for? Does this scale correctly with n? It seems like the W need to grow bigger and bigger to accommodate trace(correlation matrix) = n. Have you considered other constraints?
- Why is Gamma_n created by adding 1 to Theta_n? Why is the adversarial perturbation a matrix of 1s and not e.g. a noise vector? Also, 1_P is not defined in the notation, but J_P is. Is 1_P == J_P or does it mean something else? Furthermore, adding anything to Theta without care or renormalization takes it away from the set of feasible correlation matrices.
- Eq (5) necessarily creates non-symmetric matrices if U^s is not a multiple of the identity matrix. The diagonal should be applied on both sides if anything. How is this not a problem for the algorithm?
- How are all the lambda_r set? (also: using lower-case and upper-case lambda for different things is a bit confusing. At the beginning I thought that lower-case lambda were the diagonal entries of upper-case lambda)
- Is the W_n in formula (1) a typo and should be W_j?

Experiment questions and comments
The fact that the proposed method works better under the evaluated metrics than the same method without site information is the absolute minimum bar: If it doesn’t do that, you need to look for another method. The more interesting comparisons are ones that compare to existing literature. While there is a paragraph stating that this is somehow impossible, I do not believe that to be the case.
Two things should be done:  1) known preprocessing methods (the ones cited) should be used to remove confounds, followed by the extraction of the hierarchical decomposition 2) Other methods for correlation-matrix extraction should be used in combination with deconfounding methods, e.g. group sparse inverse covariance methods.
If evaluation ever becomes a problem, here is a suggested common evaluation metric: Generate BOLD time series. For example, given the BOLD time series in some ROIs or maybe a full hemisphere, use the extracted correlation matrix (+ variances if necessary) to generate BOLD time series for the missing part of the brain, for every voxel. This evaluation method also takes into account the parcellation chosen.

Can this method, being hierarchical, support going down to correlations at the voxel-level? This would be an interesting step forward.

It would be very interesting to see this method elevated to a general confound elimination method. Sometimes subject age is a confound. It seems like this framework can accommodate this confound in the same way as site is a confound. It would be interesting to see this method predicting a pathological variable with e.g. age, sex and site as confounds. One could evaluate its robustness against choices of test data at very different propensity scores (such as evaluated in the recent https://www.biorxiv.org/content/10.1101/2021.06.16.448764v1).


Conceptual questions and comments
There seems to be some semantic fudging and absence of logical connection around the use of the term “hierarchical”. First of all, if at all applicable at the level of neurons, this does not mean that this term is applicable at the level of voxels and vice versa. More importantly, there is no requirement or indication that incorporating a hierarchical matrix factorization idea into a connectivity estimate captures any of the aforementioned structure. One way of proving that there might be a link would be to show that the particular factorization method introduced here is better than flat methods such as TV+L1-extracted regions -> correlation matrix, followed by conditional BOLD generation.

The title is phrased way too general. This paper is not about fMRI data analysis, but very specifically about connectivity estimation. At minimum this should be rephrased.



**Time Spent Reviewing:**

3

---

> ### Author Response · Authors · 2021-08-11
> **Response to Reviewer 8hgE**
>
> We thank the reviewer for their detailed comments and for suggesting interesting future directions. We address the comments below:
>
> Technical questions and comments
>
> 1. **What is the trace(Lambda) = 1 constraint for? Does this scale correctly with n? It seems like the W need to grow bigger and bigger to accommodate trace(correlation matrix) = n. Have you considered other constraints?**
>
> Did you mean trace(correlation matrix) = P (number of nodes/regions)? The trace of lambda along with constraints on W prevents multiple solutions. If we change trace(lambda) to some other constant, then columns of W would scale appropriately, but the solution's inference would remain the same. Since the solution is reproducible and interpretable, we didn't see any reason to change the constraints. At very large P, i.e., if we are doing voxel-wise analysis, there might be a problem in the gradient descent because of the large gradients, but we have not seen an issue when working with p in the range 50-200. This is an important point which we will keep in mind when working with large P.
>
> 2. **Why is $\mathbf{\Gamma}^n$ created by adding 1 to $\mathbf{\Theta}^n$? Why is the adversarial perturbation a matrix of 1s and not e.g. a noise vector? Also, $\mathbf{1}_P$ is not defined in the notation, but $\mathbf{J}_P$ is. Is $\mathbf{1}_P == \mathbf{J}_P$ or does it mean something else? Furthermore, adding anything to $\mathbf{\Theta}$ without care or renormalization takes it away from the set of feasible correlation matrices.**
>
> Thank you for pointing this out. Here, $\mathbf{1}_p$ is a matrix containing $1$s. We consider this perturbation as rank one perturbation which will transform eigenvalues of each subject’s correlation matrix differently. Since the addition of ones does not change the symmetric property of the matrix, we just scale the matrix such that the diagonal matrix contains 1 and by Weyl's inequality about perturbation [1] it will be positive definite.  We have experimented with randomly positively scaled rank one perturbation, but the results were worse than the vanilla hSCP. Also, using rank one perturbation, it is easier to control eigenvalues (and noise added) of the modified matrix than using rank k perturbation. Hence, we decided to use the current rank one perturbation.  We will update the paper with the above details to make this section clear.
>
> 3.  **Non-symmetric matrices generated due to $\mathbf{U}^s$:**
>
> We thank the reviewer for raising an interesting point. We do not believe that is true, a counterexample would be-
> $$
> \mathbf{U} = \left(\begin{array}{cc} 1 & 0\\\\ 0 & 2 \end{array}\right) \quad \mathbf{V}=\left(\begin{array}{cc} a & b\\\\2b & c \end{array}\right)
> $$
>
> In terms of $\mathbf{U}$ and $\mathbf{V}$, the optimization problem is $||\mathbf{Z} - \mathbf{UV} ||_F^2$, where $\mathbf{Z}$ is a symmetric matrix. The gradient descent will force $\mathbf{U}$ and $\mathbf{V}$ to be such that $\mathbf{UV}$ is symmetric in order to minimize the objective function since equal weight is given to each element of the matrix on either side of the matrix. With enough degrees of freedom, the optima $\mathbf{U}$ and $\mathbf{V}$ will make the $\mathbf{UV}$ matrix symmetric. Also, we see this empirically at the convergence of the algorithm, we will update the manuscript with the visualization of $\mathbf{UV}$ matrix and visualization of $\mathbf{UV}-\mathbf{(UV)}^\top$ matrix. A $\mathbf{UVU}$ type decomposition would also work here, but there is no reason to believe that the current decomposition won't give us a symmetric matrix.
>
> 4. **How are all the $\lambda_r$ set?**
>
> Thanks for the suggestion. In the final version, we will change $\lambda_r$ to a different symbol to avoid any confusion. We select lambda from $(10^{[-3:-1]})$, for which we get the highest split sample reproducibility of the components. For very sparse and dense components, the reproducibility would be low, and there would be an optimum point at which the components would have maximum reproducibility. We will update the paper with more details about the selection of lambda values.
>
> 5. **Is the $\mathbf{W}_n$ in formula (1) a typo and should be $\mathbf{W}_j$?**
>
> Thanks for pointing it out. Yes, there is a typo; the $\mathbf{W}_n$ in formula (1) should be $\mathbf{W}_j$. We will fix the typo in the updated paper.
>
> 6. **Alternative experimental evaluation**
>
> The deconfouding approach, such as COMBAT, works at the feature level and does not take the structure of the complete feature set into account. If we directly use COMBAT on the correlation matrix, then the final matrix is not necessarily positive semi-definite because the approach focuses on removing site effects from each feature of the matrix; it does not consider that the matrix is positive semi-definite. For similar reasons, COMBAT can not be directly applied to time series; if applied, it can change the inference derived from the correlation matrix. Moreover, we are not trying to estimate a robust connectivity matrix or correlation matrix, but robust components given the correlation matrices. We believe that robust estimation of the connectivity matrix is a separate research topic. A straightforward approach would be to approximate a correlation matrix using full rank decomposition and then reduce the site effects similarly to our approach. After the reduction of site effects, the correlation matrix can be regenerated from the remaining variance.
>
> Instead, we added a baseline approach suggested by Reviewer 4. We first ran hSCP and used ComBat on the extracted $\mathbf{\Lambda}^n$ to get harmonized subject specific information $\mathbf{\Delta}^n$ for each subject. We then re-fitted $\mathbf{W}_1$ and $\mathbf{W}_2$ matrices using the below decomposition-
>
> $$\mathbf{\Theta}^n \approx \mathbf{W}_1 (\mathbf{\Delta}^n_1+\mathbf{S}_1) \mathbf{W}_1^\top \approx \mathbf{W}_1 \mathbf{W}_2(\mathbf{\Delta}^n_2+\mathbf{S}_2) (\mathbf{W}_1\mathbf{W}_2)^\top $$
>
> We added a diagonal shift matrix $\mathbf{S}$ such that $\mathbf{\Delta}^n+\mathbf{S}$ is positive for each subject and performed the optimization to estimate $\mathbf{W}$ and $\mathbf{S}$. Below is the split sample reproducibility on real dataset for $k_2 = 4$ and varied values of $k_1$
>
> $$k_1 = 10  :  0.673\pm0.049 \quad k_1 = 10 :   0.641\pm0.051  \quad k_1 = 20 : 0.639\pm0.031 \quad k_1 = 25 :  0.611\pm0.038$$,
>
> and leave one site out reproducibility on the real dataset for $k_2 = 4$ and varied values of $k_1$
>
> $$k_1 = 10  : 0.614\pm0.042 \quad k_1 = 15  :  0.594\pm0.035  \quad  k_1 = 20 :  0.542\pm0.041 \quad k_1 = 125 :  0.528\pm0.039$$
>
> From the above results, we can see that this baseline performs worse than vanilla hSCP. One reason for this might be that the harmonized lambda extracted using ComBat might not necessarily result in optimal $\mathbf{W}$, which is highly reproducible. This result bolsters our method that we need a joint optimization procedure to obtain W and lambda with reduced site effects. We have similar results for the simulated dataset as well. We update our manuscript with these experiments.
>
> 7. **Can this method, being hierarchical, support going down to correlations at the voxel-level?**
>
> This would be an interesting step forward.
> We thank the reviewer for suggesting an interesting direction. The method is not specific to particular parcellation; it can be used at voxel-level but with an additional computational cost of $O (P^2)$, as mentioned in the original hSCP paper.
>
> 8. **General confound elimination method:**
>
> Thank you for giving such an exciting direction!! We believe that the method can be used as a general confound elimination method and can be used similarly as in the suggested paper.
>
> 9. **Conceptual questions:**
>
> We agree with the reviewer. Our assumption is that spontaneous activity arises from the concerted activity of multiple components that encode system-level function, similar to sparse codes that are present at the level of neurons, information is encoded by a small number of synchronous neurons that are selective to a particular property of the stimulus. Each SCP consists of a small set of spatially distributed, functionally synchronous brain regions, forming a basic pattern of co-activation. These SCPs capture the range of resting functional connectivity patterns in the brain, although they do not necessarily need to be present in each individual or subsets of individuals. We consider hSCP and other data-driven decompositions as methods that simply produce components of brain regions that statistically appear to have relatively more synchronous activation patterns, and hence could be presumed to be parts of underlying brain networks. hSCP is a hierarchical dimensionality reduction method possessing certain desirable properties (as discussed in the paper), which not only allows us to investigate hierarchically organized co-activated brain regions, but also to probe heterogeneity of brain activity patterns across individuals. In that case, the optimal or most interpretable solution might be obtained at one scale rather than another, which is another added advantage of hSCP over single scale methods. Our results and original hSCP papers results show that the hSCP is able to capture "functional polymorphism" where similar components are present at the fine level to capture the functional heterogeneity in the data. They combine to form coarse-level components, capturing heterogeneity at a different scale that fine-level components cannot capture. We will make this part clear in the introduction of the revised paper.
>
> 10. **The title is phrased way too general.**
>
> Thank you for the suggestion. We are thinking of changing the title to 'Reducing Site Effects in Hierarchical Connectivity Patterns in Multi-site fMRI data' which directly aligns with the goal of the paper.
>
> [1] G.W. Stewart, J.G. Sun, Matrix Perturbation Theory, Academic Press

---

> > ### Comment · Reviewer_8hgE · 2021-09-02
> >
> > Thanks to the authors for this detailed response. It addresses many of my concerns and I am happy to raise my score to 6.

---

### Official Review · Reviewer_u9U8 · 2021-07-20

**Rating:** 6
**Confidence:** 4

**Summary:**

This paper proposes a combined technique based on matrix factorization and adversarial models to reduce the batch effect problem in fMRI data. The new model, called rshSCP, is designed to be robust to the site-effects in estimation of sparse connectivity pattern components. The idea is to decompose correlation matrices into a set of shared hierarchical pattern of subject components as well as site related components. Then an existing adversarial model is applied in order to increase the reproducibility of them. They also applied a simple initialization method for optimization procedure and evaluated their approach on both simulated and real datasets.

**Limitations And Societal Impact:**

The authors adequately addressed the limitations of the work under either future directions or the weaknesses of the model in the conclusion section. They also provided discussions on different challenges along with the paper and tried to alleviate the corresponding limitations by suggesting and implementing new propositions.

**Main Review:**

I found the paper is well written, the proposed idea interesting, and the motivation is clear. I enjoyed that the proposed approach uses matrix factorization to tackle the site-related batch effect problem in fMRI data. However, the base of the formulations for either the matrix factorization or adversarial learning of hSCP is already introduced in other works. The original novelty of the paper includes decomposing the correlation matrix into both hSCP and site components.

One primary concern regarding this method could be the reproducibility problem and the lack of theoretical guarantees on sub-optimal solutions. Given that this sparse hierarchical pattern is supposed to estimate a proper human brain functionality using fMRI data, it is valuable to design convex objective functions that lead to a unique optimal point rather than multiple sub-optimal solutions. In the current paradigm, the optimal solution for $W$, $D$, $U$, and $V$ is dependent on the initialization procedure as is also mentioned in the paper. Despite the careful initialization of matrices, comparing the performance of the results in either rshSCP and Adv. rshSCP, with and without random initialization, does not show significant improvement (Table 1 and Table 2).
Moreover, based on the results demonstrated in the experiment section, more justification is required for higher computational cost in the proposed method, given the slight performance improvement comparing to baselines.

A couple of further comments:
1) Try to define the dimensionality of matrices explicitly as soon as you introduce them. I am not sure what are the dimensionalities of $U^s$ and $V$? How are they defined? What do their dimensionalities represent, given that we want $U^s V$ to fit the dimensionality of $\theta^n$?
2) Designing a [group] regularizer that can capture the biological/functional feasibility of subject-specific components or site-related properties might help with non-convex optimization problems and reproducibility issues.

Overall, I feel like the contribution in this paper is not of great enough significance for this venue right now. Yet, I would be open to increasing my overall score based on the authors' responses since the idea behind the paper is interesting.

**Time Spent Reviewing:**

14

---

> ### Author Response · Authors · 2021-08-11
> **Response to Reviewer u9U8**
>
> We thank the reviewer for their detailed comments and for finding the idea of the paper interesting. We address the comments below:
>
> 1. **Optimization problem and reproducibility:**
>
> We agree with the reviewer that designing a convex problem is highly desirable, but frequently not feasible in complex, biological modeling formulations. We believe that this is not a critical issue in the current analysis because we demonstrate reproducible and biologically interpretable solutions. Also, the initialization procedure is used to make the algorithm deterministic, and with the svd-initialization procedure, the algorithm reaches convergence faster [1]. Below is some empirical evidence that supports our reasoning:
>
> (a) Our qualitative results on a real dataset show that the method estimates reproducible components, and on the simulated dataset, we show that the method can extract the components reproducibly and accurately. If we look at the qualitative aspects of the result, we see that these components are cleaner than hSCP and interpretable. We have also discussed their age-related biological significance in section 3.3 and supplementary material. A detailed discussion of the biological interpretability of components extracted can be found in [1,2].\
> (b) The hSCP method at a single level has been shown to prove effective for predicting clinical severity [3] and has been used to understand the functioning of advanced brain aging in the paper [4].\
> (c) Moreover, similar deep matrix factorization methods have been used to identify networks in fMRI data [5] and hidden representation in human faces [6], and have empirically succeeded. With the recent theoretical work in deep linear networks [7] showing that all local minima are global in deep linear networks, we believe a similar result might hold for deep matrix factorization approaches used in fMRI data analysis. We believe that in the current, we achieve a result close to optimum, which can only be verified through reproducibility experiments as of now.
>
> Theoretically analyzing the optimality of the algorithm would strongly validate the model but is beyond the scope of the current paper.
>
> 2. **Justification for the higher computational cost:**
>
> Computational cost is an important factor to consider, but more importantly in our setting is the interpretability, reproducibility, and removing the variability in the estimated components. Our results demonstrate that we are able to achieve quotative and qualitative improvement. Without reducing the site effects, we can lose the interpretability and might make incorrect inferences in the downstream analysis. Also, these inferences are not required in real-time; thus, the computational cost is of secondary importance. If we look at the running time of the hSCP, it takes 10 minutes to estimate the components on a single core of the intel i-7 machine, and adv. rshSCP takes about 20 minutes which is practically not very significant.
>
>
> 3. **Try to define the dimensionality of matrices explicitly as soon as you introduce them. I am not sure what are the dimensionalities of $\mathbf{U}^s$ and $\mathbf{V}$? How are they defined? What do their dimensionalities represent, given that we want $\mathbf{U}^s\mathbf{V}$ to fit the dimensionality of $\mathbf{\Theta}^n$?**
>
> Thanks for pointing it out. $\mathbf{U}^s$ is a real diagonal matrix of size $P \times P$, and $\mathbf{V}$ is a real matrix of size $P \times P$, thus multiplying $\mathbf{U}^s$ and $\mathbf{V}$ will result in a $P \times P$ matrix matching the dimensionality of $\mathbf{\Theta}^n$. We hypothesize that $\mathbf{V}$ stores site and scanner information for all the possible available data. For each site $s$, we have space $\mathbf{U}^s$ storing site-specific information for $s=1,\ldots, S$.
>
> 4. **Significance:**
>
> The current analysis is an important step for using hSCP for large-scale analysis where the datasets are pooled from multiple sites. Robust hSCP are essential for downstream analysis such as-
>
> (a) Using these components for predicting clinical severity.\
> (b) Estimating variation in the components with age and help in the understanding reorganization of the aging human brain (partly discussed in Section 3.3 and Supplementary material section E).\
> (c) Understanding the role of functional networks in cognitive performance by finding an association between the components and various cognitive tests such as Trail Making Test, Mini-Mental State Exam, Digit symbol substitution test, etc.
>
> We can make incorrect inferences with noisy components without reducing the site variability, as shown in section 3.3. Thus reducing site effects is essential for extracting interpretable and reproducible components in pooled large-scale studies.
>
> [1] Sahoo, Dushyant, Theodore D. Satterthwaite, and Christos Davatzikos. "Hierarchical extraction of functional connectivity components in human brain using resting-state fmri." IEEE Transactions on Medical Imaging 40.3 (2020): 940-950.\
> [2] Eavani, Harini, et al. "Identifying sparse connectivity patterns in the brain using resting-state fMRI." Neuroimage 105 (2015): 286-299.\
> [3] D'Souza, Niharika Shimona, et al. "A joint network optimization framework to predict clinical severity from resting state functional MRI data." NeuroImage 206 (2020): 116314.\
> [4] Eavani, Harini, et al. "Heterogeneity of structural and functional imaging patterns of advanced brain aging revealed via machine learning methods." Neurobiology of aging 71 (2018): 41-50.\
> [5] Li, Hongming, Xiaofeng Zhu, and Yong Fan. "Identification of multi-scale hierarchical brain functional networks using deep matrix factorization." International Conference on Medical Image Computing and Computer-Assisted Intervention. Springer, Cham, 2018.\
> [6] Trigeorgis, George, et al. "A deep matrix factorization method for learning attribute representations." IEEE transactions on pattern analysis and machine intelligence 39.3 (2016): 417-429.\
> [7] Laurent, Thomas, and James Brecht. "Deep linear networks with arbitrary loss: All local minima are global." International conference on machine learning. PMLR, 2018.\

---

> > ### Comment · Reviewer_u9U8 · 2021-09-02
> > **increasing the score to 6**
> >
> > Thanks for the author's response. They addressed most of my concerns. So I would like to increase the score to 6.

---

### Decision · Program_Chairs · 2021-09-27

**Decision:**

Accept (Poster)

**Comment:**

Dear authors,

despite some initial concerns reviewers have now, after discussion and reading
your rebuttal, endorsed the paper (although not strongly). I will therefore suggest
a rather positive outcome for this work.

Best regards,
The AC